# LatentChem: From Textual CoT to Latent Thinking in Chemical Reasoning

Xinwu Ye [* 1]   Yicheng Mao [* 2]   Yuxuan Liao [3]   Jia Zhang [4]   Yimeng Liu [5]   Hao Li [2]   Fang Wu [6]   Zhiwei Li [7]
Zehong Wang [8]   Zhiyuan Liu [9]   Zhenfei Yin [10]   Li Yuan [2]   Philip Torr [10]   Huan Sun [11]   Xiangxiang Zeng [4]
Mengdi Wang [12]   Le Cong [6]   Shenghua Gao [1]   Xiangru Tang [3]

## Abstract

Current chemical large language models (LLMs) predominantly rely on explicit Chain-of-Thought (CoT) to solve complex reasoning problems. However, forcing nonverbal tacit chemical logic into discrete natural language imposes a fundamental "modality mismatch," creating an artificial bottleneck for reasoning. We introduce **LatentChem**, a reasoning interface that decouples chemical logic from linguistic generation, enabling the model to process information via continuous thought vectors and dynamic perception. Our investigation reveals a pivotal emergent behavior: spontaneous internalization, defined here as self-selected under outcome-only optimization. When optimized for task success, the model abandons verbose textual derivations in favor of implicit latent computation, suggesting that it identifies the continuous manifold as a more native substrate for chemical logic. This paradigm shift also proves to be a superior computational strategy: LatentChem achieves a **59.88%** non-tie win rate against the strong CoT baseline on the rigorous ChemCoTBench, while delivering a broad **10.84×** average reduction in reasoning step overhead (**5.96×** wall-clock speedup) across all evaluated benchmarks. Our results provide empirical evidence that chemical reasoning is more naturally and effectively realized as continuous latent dynamics rather than discretized linguistic trajectories.

---

*Equal contribution [1]The University of Hong Kong [2]Peking University [3]Yale University [4]Hunan University [5]University of Toronto [6]Stanford University [7]Hong Kong University of Science and Technology (Guangzhou) [8]University of Notre Dame [9]National University of Singapore [10]University of Oxford [11]The Ohio State University [12]Princeton University. Correspondence to: Xiangru Tang <xiangru.tang@yale.edu>.

*Proceedings of the 43rd International Conference on Machine Learning*, Seoul, South Korea. PMLR 306, 2026. Copyright 2026 by the author(s).

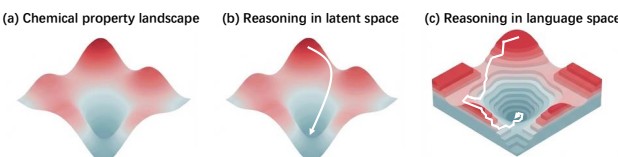

*Figure 1.* Conceptual illustration of the modality mismatch. (a) The intrinsic chemical property landscape is continuous and high-dimensional. (b) We posit that a continuous latent space can theoretically offer a smoother optimization surface akin to the property landscape, avoiding the jagged trajectories of discrete tokens. (c) Explicit CoT forces an artificial discretization of the landscape into discrete tokens. This quantization results in jagged, inefficient trajectories.

## 1. Introduction

LLMs have emerged as transformative tools for scientific discovery, facilitating tasks ranging from molecule generation to reaction synthesis. In this context, chemical reasoning is typically mediated through explicit CoT (Zhang et al., 2024; Zhao et al., 2025). In current chemical LLM paradigms, models are trained to linearize a complex physicochemical intuitions, such as electron delocalization or steric hindrance, into a discrete sequence of natural language tokens before arriving at a solution. While this CoT approach has unified scientific tasks under a generative framework, it relies on the implicit assumption that natural language is an adequate vehicle for the continuous physicochemical dynamics inherent to chemistry.

However, it remains unclear whether such discrete symbolic reasoning is optimal, for complex chemical reasoning tasks. We hypothesize that forcing chemical logic into a linguistic bottleneck results in a fundamental "modality mismatch" (Figure 1). Much like an expert chemist who manipulates abstract 3D structures mentally before verbalizing the result, we posit that the core of chemical reasoning, such as navigating manifolds, optimizing properties, and identifying substructures, is more naturally performed in a continuous latent space. In this view, natural language should serve merely as the input/output interface rather than the computational substrate for the reasoning process itself.

To study this hypothesis, we introduce LatentChem[1], a latent

---

[1]The code and model weights are publicly available at GitHub

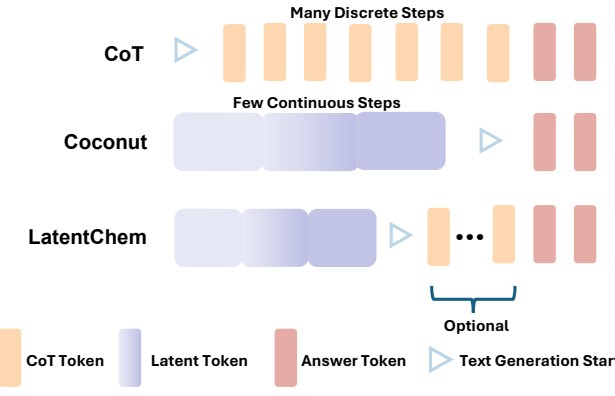

*Figure 2.* Comparison of reasoning paradigms in chemical LLMs. (Top) Explicit CoT relies on discrete linguistic steps, forcing high-dimensional chemical intuition into a constrained textual bottleneck. (Middle) Generic latent reasoning (e.g., Coconut(Hao et al., 2025c)) shifts reasoning to a continuous latent space but treats molecular embeddings as a static context, limiting the ability to focus on different substructures during reasoning. (Bottom) LatentChem (Ours) introduces a dynamic perception-reasoning loop. Via the ChemUpdater mechanism, latent thoughts actively re-query and refine the molecular representation at each step, ensuring structure-aware reasoning.

reasoning interface designed to decouple chemical reasoning from explicit natural language generation. As illustrated in Figure 2, unlike standard models that rigidly bind reasoning to token output, LatentChem injects a lightweight sequence of continuous thought vectors between perception and generation. Crucially, we equip this interface with a ChemUpdater mechanism, allowing these latent thoughts to actively re-query molecular features in a dynamic loop. This architecture serves as an experimental instrument, allowing us to observe whether the model prefers to reason via explicit text or through internal latent dynamics when optimized for task success.

Our investigation reveals an intriguing emergent behavior: the model exhibits spontaneous internalization of chemical logic. Here, spontaneous internalization refers to self-selection under outcome-only optimization, rather than objective-independent discovery of latent reasoning from scratch. Despite being initialized with explicit CoT data, when optimized via reinforcement learning with rewards only on format adherence, validity, and correctness, LatentChem voluntarily discards verbose textual reasoning chains in the majority of tasks. Instead, it predominantly collapses the reasoning process into the continuous latent space, outputting solutions directly after a sequence of "silent" thought vectors. While this shift sacrifices the explicit readability of intermediate steps, it strongly suggests that the

and HuggingFace.

model perceives natural language as a low-bandwidth constraint, identifying the continuous latent reasoning as a more native and expressive mode for chemical logic.

Crucially, this spontaneous shift does not imply that the model is merely taking a shortcut to minimize generation effort, as such a behavior would typically degrade task performance. On the contrary, our experimental results demonstrate that this internalization represents a superior computational strategy. Quantitatively, by choosing to compress linguistic steps into compact latent states, LatentChem achieves a dramatic reduction in reasoning step overhead, averaging $10.84\times$ across all benchmarks (with a measured $5.96\times$ wall-clock speedup overall). In molecule optimization and reaction tasks, this reduction factor even exceeds $28\times$. This efficiency is achieved alongside dominant performance, as evidenced by extensive evaluations across four diverse benchmarks, ChemCoTBench (Hao et al., 2025a), Mol-Instructions (Fang et al., 2024), ChEBI-20 (Edwards et al., 2021), and ChemLLMBench (Guo et al., 2023). Notably, LatentChem achieves a $59.88\%$ non-tie win rate against the strong explicit CoT baseline on the reasoning-intensive ChemCoTBench. The simultaneous improvement in both token efficiency and accuracy confirms that decoupling reasoning from language unlocks a more native and effective reasoning paradigm for chemical LLM.

In summary, our contributions are as follows:

- We establish and characterize a chemistry-specific latent reasoning paradigm relative to explicit CoT, challenging the necessity of natural language CoT for chemical tasks.
- We introduce LatentChem, a system that empowers chemical LLMs to perform latent thinking via continuous thought vectors and a dynamic perception loop.
- We report the phenomenon of spontaneous internalization, where the model absorbs explicit reasoning into the latent space, validating the efficiency of continuous representations.
- We demonstrate that this paradigm shift yields superior performance and token efficiency across diverse chemical benchmarks compared to explicit CoT approaches.

## 2. Related Work

**LLMs for chemical tasks.** While general LLMs struggle with the specialized logic required for scientific discovery (Hatakeyama-Sato et al., 2023; Jiang et al., 2025a; Sallam et al., 2024; Ye et al., 2025), the field has progressed through domain-specific fine-tuning (Zhang et al., 2024; Zhao et al., 2025), multimodal architectures (Tan et al., 2025; Jiang et al., 2025b; Li et al., 2025; Bai et al., 2025; Lv et al., 2025; Liu et al., 2023) that integrate 2D/3D geometries to capture chemical topology (Pei et al., 2024; Xia et al., 2025), and

agentic decomposition of tasks (Tang et al., 2025). Furthermore, reinforcement learning is increasingly utilized to enforce logical consistency and physical validity (Narayanan et al., 2026; Zhuang et al., 2025; Wang et al., 2025c), often within unified training frameworks (Zuo et al., 2025). Despite these algorithmic strides, current systems remain prone to hallucination (Hao et al., 2025b) and often lack the robust planning capabilities necessary for complex chemical reasoning tasks (Zhang et al., 2025).

**Latent thinking in LLMs.** An emerging paradigm shifts reasoning from explicit token generation to implicit processing within high-dimensional latent spaces, bypassing the linguistic bottleneck of sequential text. Pioneering works demonstrate that models can internalize reasoning steps by feeding hidden states directly as input and benefit from multiple reasoning paths (Deng et al., 2024; Hao et al., 2025c; Zhu et al., 2026). To structure these internal processes, research has compressed reasoning to "contemplation tokens" (Cheng & Durme, 2024) or "capsules" (Shan et al., 2025), and employs differentiable caches for "soft thoughts" and memories (Xu et al., 2025a; Liu et al., 2025). These latent states are further refined through auxiliary supervision, synthetic target for alignment, self-rewarding mechanisms, and variance optimization (Wei et al., 2026; Wang et al., 2025b; Chen et al., 2024; Wang et al., 2025a). Crucially, this nonsequential nature unlocks test-time compute scaling: models can deepen reasoning through recursive unrolling (Geiping et al., 2026; Aleksandrov et al., 2025), leverage native parallelism via Jacobi iteration, (Wen et al., 2025; Wu et al., 2025) or scaled through diverse initializations (Xu et al., 2025b). These approaches establish the foundation for the iterative, structure-aware computation that our method extends to dynamically refine molecular features.

# 3. LatentChem: A Latent Reasoning Interface

We present LatentChem, a framework designed to decouple chemical reasoning from linguistic generation. As illustrated in Figure 3, the system comprises three core components: (1) a latent thinking architecture that establishes the structural foundation for non-linguistic reasoning; (2) an active perceptual refinement mechanism that enables the model to dynamically update its molecular understanding; and (3) a progressive training protocol that evolves the model from simple alignment to outcome-driven policy optimization.

## 3.1. Latent Thinking Neural Architecture

The architecture of LatentChem introduces three specialized modules integrated into a general-purpose LLM backbone: the Chemical Adapter for initial perception, the ChemUpdater for dynamic refinement, and the latent projector for

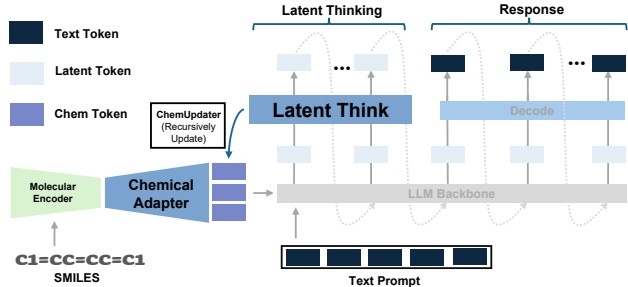

*Figure 3.* The overview of LatentChem architecture. The system decouples reasoning from generation via a dedicated latent thinking phase. (1) A chemical adapter aligns molecular features with the LLM space. (2) During the reasoning phase, the model generates a sequence of continuous latent thought vectors. (3) The ChemUpdater allows these thoughts to recursively re-query the molecular encoder, refining the representation based on the current reasoning state before the final response is decoded.

reasoning continuity.

**Chemical Adapter.** To bridge the modality gap between continuous molecular features and the discrete embedding space, we employ a query-based attention projector (specifically, a Perceiver Resampler architecture (Alayrac et al., 2022)). Given a molecule, we first extract a variable-length sequence of dense features $\mathbf{H}_{mol} \in \mathbb{R}^{L \times d_{enc}}$ using the pre-trained SMI-TED encoder (Soares et al., 2024). To characterize the molecule comprehensively, we initialize $N$ learnable latent queries $\mathbf{Q} \in \mathbb{R}^{N \times d_{llm}}$. These queries serve as semantic anchors, extracting specific chemical attributes via cross-attention:

$$\mathbf{H}_{chem} = W(\text{LN}(\mathbf{Q} + \text{MHA}(\mathbf{Q}, \mathbf{H}_{mol}, \mathbf{H}_{mol}))),$$

where $W$ projects dimensions to the LLM size $d_{llm}$. This mechanism compresses variable-length molecular information into a fixed number of "ChemTokens" $\mathbf{H}_{chem}$, which are prepended to the textual instruction embeddings as a prefix soft prompt.

**ChemUpdater and Latent Projector.** To enable dynamic interaction between reasoning and perception, we incorporate two complementary modules. First, unlike standard models that treat encoders as static feature extractors, the ChemUpdater enables dynamic perceptual refinement. It utilizes a cross-attention layer where the current ChemTokens $\mathbf{H}_{chem}^{(t)}$ serve as queries and the accumulated reasoning history $\mathbf{Z}_{1:t}$ serves as keys and values, allowing the model to shift its focus on molecular substructures as reasoning evolves. Second, to close the loop, we employ a latent projector, implemented as a lightweight residual feed-forward network (FFN). This module maps the raw hidden state $\mathbf{z}_t$ output by the LLM backbone back into the input embedding space, creating the continuous input vector required for

the subsequent step and effectively bypassing the discrete tokenization bottleneck.

### 3.2. Latent Thinking with Active Perceptual Refinement

The core of LatentChem is the continuous latent thinking loop, which transforms the generation process into a dual-pathway system of thought generation and active perceptual updating. Operationally, latent thinking denotes recurrent generation of continuous hidden states that are fed back into the model without textual decoding; one latent step denotes one such recurrent update; and continuous reasoning denotes intermediate computation in latent space rather than through discrete text tokens. At each reasoning step $t$, the LLM produces a high-dimensional hidden state $\mathbf{z}_t$. Instead of decoding this state into a text token, the model treats it as a "thought vector" that drives two parallel processes.

On the perceptual pathway, the raw state $\mathbf{z}_t$ is appended to the history $\mathbf{Z}_{1:t}$ to trigger the ChemUpdater. The model actively refreshes its molecular representation via:

$$\mathbf{H}_{chem}^{(t+1)} = \mathrm{LN}\Big(\mathbf{H}_{chem}^{(t)} + \mathrm{CrossAttn}(\mathbf{H}_{chem}^{(t)}, \mathbf{Z}_{1:t}, \mathbf{Z}_{1:t})\Big).$$

This ensures the model does not just "see" a static structure but continuously refines its focus based on the latest thought. Simultaneously, on the reasoning pathway, the state $\mathbf{z}_t$ is processed by the latent projector to form the input for the next step: $\mathbf{h}_{t+1} = \mathbf{z}_t + \mathrm{FFN}(\mathrm{LN}(\mathbf{z}_t))$. The projected vector $\mathbf{h}_{t+1}$ is fed back into the backbone LLM to trigger $\mathbf{z}_{t+1}$. This cycle continues until the model predicts a discrete termination token `<end_latent>` or reaches a budget limit $T_{max}$, at which point the final refined context is used to decode the explicit response.

### 3.3. Training Protocol: From Explicit Alignment to Implicit Reasoning

We design a progressive four-stage training protocol to instill latent reasoning capabilities (Table 1). The first three stages focus on supervised structural alignment and mind activation, establishing the prerequisites for reasoning. Crucially, the final stage employs Group Relative Policy Optimization (GRPO) (Shao et al., 2024), a reinforcement learning paradigm that allows the model to autonomously explore and optimize its reasoning trajectory based on task feedback. We provide an expanded elaboration of the protocol in Appendix C and detailed implementation hyperparameters in Appendix D.

**Training data.** Our model is trained on ChemCoTDataset (Hao et al., 2025a), a specialized corpus designed for step-by-step chemical reasoning. We utilize the snapshot from

*Table 1.* Overview of training strategies. We detail the module status and supervision signals across four stages.

| Stage | Module | | Supervision | | |
|---|---|---|---|---|---|
| | Adapter + LLM | Latent Thinking | CoT | Answer | Reward |
| 1 | Trainable | Disabled | ✗ | ✓ | ✗ |
| 2 | Trainable | Disabled | ✓ | ✓ | ✗ |
| 3 | Frozen | Trainable | ✓ | ✓ | ✗ |
| 4 | Trainable | Frozen | ✗ | ✗ | ✓ |

November 2025,[2] which comprises approximately 14k high-quality CoT-annotated samples covering diverse tasks such as molecular understanding, editing, optimization, and reaction prediction. The per-task distribution follows the official ChemCoTDataset release.

**Stage 1: Establishing the molecular-linguistic mapping.** The primary goal is to align the projected ChemTokens $\mathbf{H}_{chem}$ with the LLM's semantic space. We use "answer-only" supervision, suppressing intermediate reasoning steps to force the chemical adapter to compress all necessary properties into the ChemTokens. To prevent textual overfitting, we employ a counterfactual alignment strategy, optimizing a hinge loss $\mathcal{L}_{CF}$ that maximizes the margin between the likelihood of the answer given clean versus perturbed molecular inputs. The total objective is $\mathcal{L}_{total}^{(1)} = \mathcal{L}_{clean} + \lambda\mathcal{L}_{CF}$.

**Stage 2: SFT for molecule-aware CoT.** We then transition to full sequence training, requiring explicit CoT derivations $\mathbf{y}_{cot}$ prior to the answer $\mathbf{y}_{ans}$. To ensure the generated reasoning is strictly grounded in molecular structure rather than textual priors, we extend the counterfactual alignment strategy to the entire sequence $\mathbf{y}_{full} = [\mathbf{y}_{cot}, \mathbf{y}_{ans}]$.

**Stage 3: Chemistry-aware latent mind activation.** In this phase, we activate the latent thinking modules (updater and projector). To prevent the massive LLM backbone from treating the lightweight latent signals as noise, we freeze both the Chemical Adapter and the LLM backbone, updating only the updater and projector. This constraint forces the latent modules to adapt to the frozen semantic space of the LLM, compelling them to generate "legible" thought vectors that guide the fixed decoder toward the correct CoT path.

**Stage 4: GRPO with latent thinking budget.** With the latent mind activated, we employ GRPO to refine the model's policy. Reversing the freezing strategy from the previous stage, we now freeze the latent thinking modules and fine-tune the remaining parameters. This strategy treats the

---

[2] The dataset was accessed on November 25, 2025. Since ChemCoTDataset is continuously updated, we explicitly state the access date to ensure reproducibility.

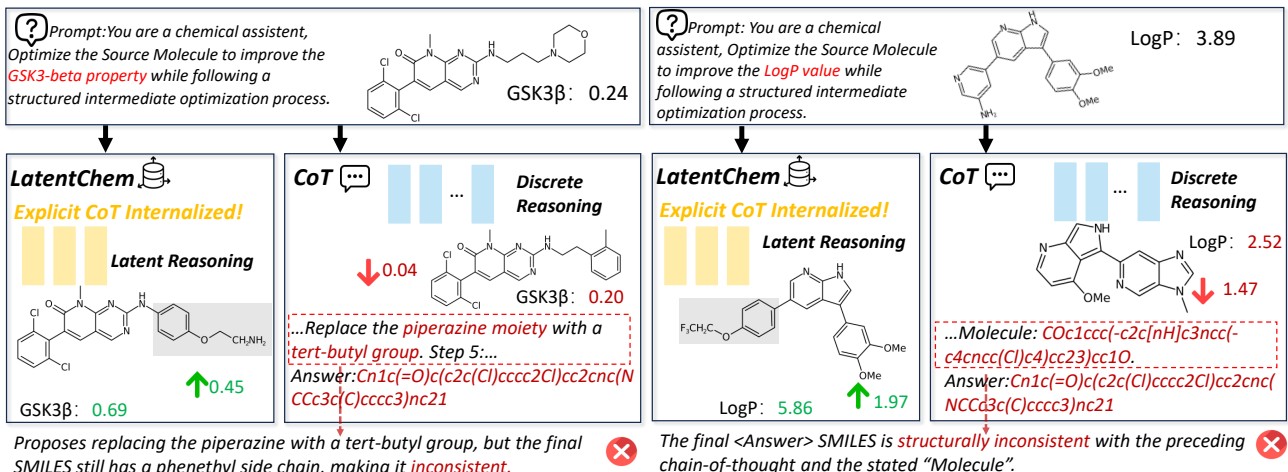

*Figure 4.* Case study: spontaneous internalization vs. explicit CoT. While the standard CoT model generates verbose reasoning chains that fail to execute the planned modification, LatentChem spontaneously internalizes these logics. It bypasses textual output, utilizing the latent space to perform structural optimization.

learned latent dynamics as a stable "internal simulator," optimizing the backbone LLM to effectively utilize these latent thoughts for decision-making. The optimization is guided by a composite reward function comprising three terms: format adherence, answer validity, and answer correctness. The reward contains no term for brevity or CoT omission. Critically, by removing the supervision for explicit CoT generation and rewarding only the final result, we enable the model to explore the most efficient reasoning pathway.

# 4. Emergent Properties of Latent Chemical Reasoning

Having established the LatentChem interface, we now investigate the nature of the reasoning process that emerges from the training protocol. To dissect these behaviors, we utilize ChemCoTBench (Hao et al., 2025a) as our primary testbed. Specifically, our analysis focuses on two representative tasks: molecule optimization and molecule understanding. Unless otherwise stated, all comparisons in this section are conducted between LatentChem and the explicit CoT baseline (Stage 1+2+4).

## 4.1. Spontaneous CoT Internalization

A pivotal discovery in our experiments is the emergent behavior observed during the transition from SFT to reinforcement learning (Stage 4). Despite being heavily supervised with explicit textual CoT data in before Stage 4, LatentChem internalized the entirety of explicit textual reasoning after GRPO training.

**The phenomenon.** Remarkably, despite being optimized using rewards constrained only by format adherence, valid-

ity, and correctness, the model ceased generating intermediate textual steps. Instead, the generation process follows a distinct pattern: the model performs a sequence of latent thinking steps, outputs a trivial artifact (typically a single "." or ":") as a transition token, and immediately generates the target XML tags. Two cases are presented in Figure 4, where LatentChem bypasses the verbose and often hallucinated textual plan generated by the CoT baseline, utilizing the latent space to perform accurate structural optimization.

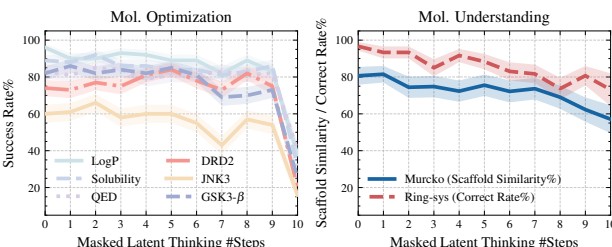

*Figure 5.* Causal necessity analysis. We measure task performance (success rate, scaffold similarity, or correct rate) as the first $k$ latent tokens are replaced with Gaussian noise. The overall degradation trend observed in both Molecule Optimization (left) and Understanding (right) confirms that early latent states encode critical precursors for the final solution rather than redundant noise. Error bars denote $\pm 1$ standard error.

**Causal necessity verification.** To confirm that this "silent" phase performs actual computation rather than acting as a passive delay, we conduct causal ablation experiments. As shown in Figure 5, replacing the initial latent steps with Gaussian noise leads to an overall performance degradation trend. This rigorously establishes that the internalized latent states encode critical structural precursors and are

functionally essential for the final generation.

**Insight: optimization over imitation.** This internalization was not explicitly enforced by any loss penalty on output length. Rather, it represents a strategy shift driven purely by the reward signal. This observation serves as powerful empirical validation for our core hypothesis regarding the "modality mismatch." It suggests that when the model is granted the freedom to reason for chemical tasks, it identifies the high-dimensional, continuous latent space as a more native and structurally consistent workspace than the discrete, low-bandwidth channel of natural language.

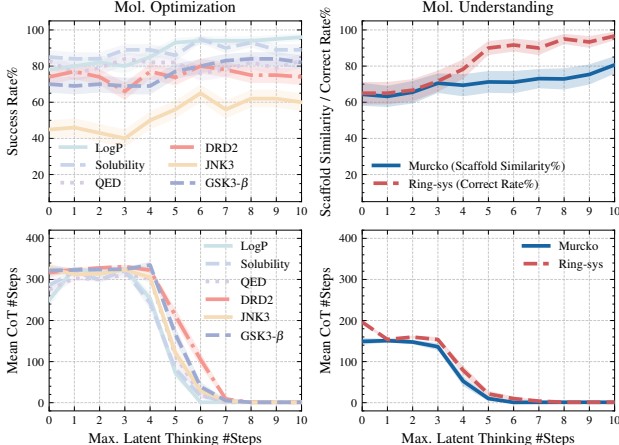

*Figure 6.* Budget stress testing. We jointly monitor generated explicit CoT length and task performance as the maximum allowed latent thinking steps decreases. As the latent budget is restricted, the model externalizes reasoning into text, while the performance panel shows how task success changes across budgets.

### 4.2. Emergent Compensatory Dynamics and Causal Necessity

Building on the discovery of CoT internalization, a critical question arises: *is this behavior a rigid pattern or a flexible strategy?* To investigate this, we conduct budget stress testing on the molecule optimization and understanding tasks, monitoring the length of generated explicit CoT as the maximum allowed latent thinking steps decrease. We also track task performance under the same budget sweep, so Figure 6 shows both the compensatory change in reasoning format and the corresponding performance response.

As illustrated in Figure 6, we observe a remarkable hydraulic trade-off. When the latent budget is sufficient (*e.g.,* $T \geq 6$), the model maintains the "internalized" state with near-zero CoT length. However, as we progressively starve the model of latent thinking steps ($T < 6$), it reactivates explicit CoT without any external instruction. This dynamic behavior suggests that the model treats implicit and explicit reasoning as communicating vessels, revealing that LatentChem has

learned a robust mechanism for arbitrating between implicit and explicit reasoning pathways based on its computational capacity.

### 4.3. Unveiling the Latent Manifold

To investigate the internal dynamics of the latent thinking process, we examine the evolutionary trajectories of Chem-Tokens across different optimization tasks. We first project the high-dimensional latent states into a 2D space using t-SNE to visualize the global distribution shift. As illustrated in Figure 7(a), the dynamics reveal a phenomenon of immediate functional partitioning rather than gradual evolution. Initially, at Step 0, representations from all tasks are densely entangled, indicating a generic, task-agnostic encoding. However, within the first two latent steps, the trajectories undergo a dramatic divergence, rapidly snapping into distinct, task-specific clusters. From Step 3 to 10, these spatial configurations remain stable. This rapid convergence suggests that the model possesses an intrinsic efficiency; it does not require a prolonged period to locate the optimization manifold but instead reconfigures the molecular representation almost instantaneously to align with the specific chemical goal.

To determine whether these drastic latent updates compromise the physical validity of the representation, we further employ representational similarity analysis (RSA) to quantify the alignment between the latent geometry and the true chemical topology. Specifically, we track the Spearman correlation between the pairwise cosine similarity of Chem-Tokens and the Tanimoto similarity of their corresponding molecular fingerprints. As shown in Figure 7(b), the structural correlation remains remarkably stable across all reasoning steps. It implies that the massive updates occurring in the first few steps are orthogonal to the direction of structural information. The latent thinking process acts as a lossless reservoir of structural information throughout the reasoning chain.

## 5. Benchmarking Latent vs. Explicit Paradigms

To substantiate that the observed latent reasoning dynamics translate into superior performance, we conduct comprehensive evaluations across four diverse chemical benchmarks.

### 5.1. Experimental Setup

**Task taxonomy.** We categorize the target capabilities into two paradigms based on the solution space. Open-ended generative tasks (*e.g.,* molecule optimization) lack a unique optimal solution, requiring the model to explore the chemical manifold to generate valid candidates. In contrast, pre-

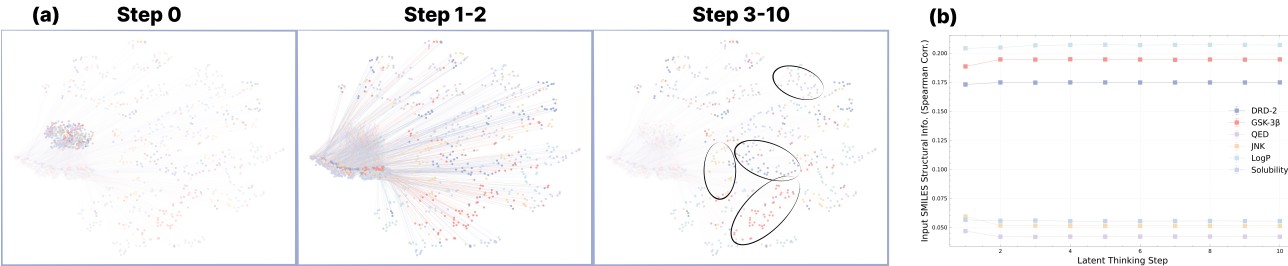

*Figure 7.* Visualization of latent dynamics during the reasoning process. (a) t-SNE projections of ChemTokens at different steps show a transition from initial entanglement to rapid disentanglement. The representations diverge significantly within the first two steps to form task-specific clusters and remain stable thereafter. (b) RSA results showing the correlation between latent geometry and chemical topology. The flat trajectories indicate that despite the significant spatial updates for task adaptation, the model preserves structural fidelity throughout the reasoning chain, suggesting that optimization occurs orthogonally to topological encoding.

cise closed-ended tasks (*e.g.,* reaction prediction) demand unique, deterministic answers, functioning as rigorous mapping problems. We leverage this dichotomy to investigate the differential impact of latent reasoning on creative exploration versus deterministic execution.

**Benchmarks.** We evaluate our method on four established benchmarks: ChemCoTBench (Hao et al., 2025a), Mol-Instructions (Fang et al., 2024), ChEBI-20 (Edwards et al., 2021), and ChemLLMBench (Guo et al., 2023). To rigorously assess the molecular reasoning and generation capabilities targeted by our framework, the evaluation suite is curated to focus on chemical competencies aligned with the instruction-tuning phase. Consequently, for Chem-CoTBench, the evaluation covers 1,120 samples covering molecule optimization, understanding, editing, and reaction prediction. The molecule-oriented component of Mol-Instructions provides 4,000 samples across forward reaction prediction, retrosynthesis, reagent prediction, and description generation. ChEBI-20 contributes its standard test set of 3,297 molecule-description pairs. Finally, 600 samples from ChemLLMBench are included, spanning seven relevant subtasks, including molecule captioning and various reaction prediction challenges, to verify performance consistency across diverse evaluation sources.

**Baselines.** We compare LatentChem against three categories of baselines. All molecule-aware models (LatentChem, explicit chemical LLMs, and Coconut-Chem) share the identical Qwen-3-8B (Yang et al., 2025) backbone equipped with the SMI-TED encoder and Chemical Adapter. The categories include: (1) text-only LLMs (Qwen-3-8B and Qwen-3-8B SFT) which establish a performance floor; (2) explicit chemical LLMs, which utilize the exact same Chemical Adapter+Qwen-3-8B neural architecture as LatentChem but are constrained to standard CoT reasoning (reported at Stage 1, Stage 1+2, and the GRPO-optimized Stage 1+2+4); Stage 1+2+4 is therefore the explicit-CoT baseline and contains no latent modules. and (3) generic

latent models, specifically Coconut-Chem, which adapts the Coconut paradigm (Hao et al., 2025c) to our encoder-adapter setup.

**Metrics.** We employ diverse task-specific metrics (detailed in Appendix E). To facilitate a rigorous statistical comparison, we employ the sign test (Dixon & Mood, 1946). Aligning with this framework's theoretical premise that ties are non-informative, we quantify performance using the non-tie win rate ($\mathcal{R}^*_{win}$), which is computed as: $\mathcal{R}^*_{win} = \frac{N_{win}}{N_{win} + N_{loss}}$.

### 5.2. Main Results

**Generative capabilities.** The advantages of the latent paradigm are most pronounced in open-ended generative tasks (Table 2), surpassing all baselines on 14 out of 15 metrics. In molecule optimization, reasoning in the continuous latent manifold establishes a substantial lead over discrete planning. Furthermore, the 8B-parameter LatentChem consistently outperforms the state-of-the-art results from Claude 3.7 Sonnet reported in ChemCoTBench (see Appendix J for detailed comparison). Notably, on the challenging GSK3-$\beta$ task, LatentChem achieves a success rate of **82%** (vs. 67% for explicit CoT). This validates that high-dimensional latent exploration unlocks a level of "generative creativity" and navigational precision that is inaccessible to explicit token-based methods.

**Robustness on deterministic mappings.** The performance contrast between task categories offers critical insight into the mechanism of latent thinking. As shown in the "Closed" columns of Table 3, while the latent approach maintains a leading position, the margin is narrower compared to open-ended tasks. This aligns with the nature of closed-ended tasks as deterministic mapping problems, where the benefits of creative exploration are naturally saturated. Crucially, however, internalized reasoning preserves foundational precision: although it does not yield per-subtask dom-

*Table 2.* Detailed performance on open-ended generative tasks. Metrics include property improvement ($\Delta \uparrow$) and success rate (SR%$\uparrow$) for molecule optimization, alongside METEOR$\uparrow$ scores for description. Results for closed-ended tasks are detailed in Appendices F.

| Category | Method | Molecule Optimization (ChemCoTBench) | | | | | | | | | | | | Molecule Description | | |
| | | LogP | | Solubility | | QED | | DRD2 | | JNK3 | | GSK3-$\beta$ | | Mol-Instr. | ChEBI-20 | ChemLLM. |
| | | $\Delta$ | SR% | $\Delta$ | SR% | $\Delta$ | SR% | $\Delta$ | SR% | $\Delta$ | SR% | $\Delta$ | SR% | METEOR | METEOR | METEOR |
|---|---|---|---|---|---|---|---|---|---|---|---|---|---|---|---|---|
| Text-only | Qwen-3-8B | 0.00 | 3 | 0.00 | 4 | 0.00 | 4 | 0.00 | 4 | -0.01 | 0 | 0.00 | 2 | 0.09 | 0.12 | 0.09 |
| | Qwen-3-8B (SFT) | 0.15 | 47 | 0.48 | 52 | 0.10 | 48 | 0.04 | 38 | -0.02 | 20 | 0.02 | 36 | **0.12** | 0.12 | 0.13 |
| Chem. LLMs | Stage 1 | 0.35 | 61 | 0.87 | 78 | 0.14 | 73 | 0.20 | 66 | 0.02 | 40 | 0.13 | 61 | 0.10 | 0.08 | 0.11 |
| | Stage 1+2 | 0.51 | 60 | 0.82 | 69 | 0.14 | 72 | 0.12 | 53 | 0.01 | 32 | 0.08 | 39 | 0.07 | 0.05 | 0.06 |
| | Stage 1+2+4 | 0.67 | 77 | 1.17 | 86 | **0.19** | 82 | 0.19 | 70 | 0.03 | 44 | 0.13 | 67 | 0.07 | 0.05 | 0.07 |
| Latent Chem. LLMs | Coconut-Chem | 0.17 | 44 | 0.57 | 58 | 0.15 | 67 | 0.04 | 45 | -0.00 | 35 | 0.06 | 47 | 0.09 | 0.07 | 0.04 |
| | LatentChem (Ours) | **1.37** | **96** | **1.53** | **89** | 0.18 | **83** | **0.26** | **74** | **0.08** | **60** | **0.17** | **82** | **0.12** | **0.15** | **0.16** |

*Table 3.* Main results on chemical benchmarks. We report the non-tie win rate ($\mathcal{R}^*_{win} \uparrow$) of each method compared to the reference CoT baseline (Stage 1+2+4). The "All" column denotes the aggregated win rate on each benchmark. Bold and underlined values indicate the best and second-best performance, respectively. * denotes statistical significance with $p < 0.01$ according to the sign test.

| Category | Method | ChemCoTBench | | | Mol-Instructions | | | ChemLLMBench | | | ChEBI-20 |
| | | Open | Closed | All | Open | Closed | All | Open | Closed | All | Open |
|---|---|---|---|---|---|---|---|---|---|---|---|
| Text-only | Qwen-3-8B | 13.98* | 2.80* | 8.61* | 58.65* | 21.87* | 23.97* | 66.67* | 19.58* | 22.78* | 74.86* |
| | Qwen-3-8B (SFT) | 28.08* | 20.63* | 25.75* | **73.13*** | 35.92* | 38.39* | 83.14* | 37.51* | 41.11* | 80.85* |
| Chem. LLMs | Stage 1 | 43.84* | 36.76* | 41.73* | 63.00* | **50.49** | **51.25*** | 68.47* | 50.56 | 51.57 | 64.80* |
| | Stage 1+2 | 38.35* | 28.44* | 35.50* | 50.36 | 43.69* | 44.10* | 59.77 | 45.12* | 46.21* | 54.60* |
| Latent Chem. LLMs | Coconut-Chem | 34.71* | 26.32* | 32.11* | 57.94* | 39.45* | 40.57* | 75.00* | 42.46* | 44.82* | 65.58* |
| | LatentChem (Ours) | **63.73*** | **50.00** | **59.88*** | 72.00* | 48.36* | 49.88 | **87.37*** | **52.87** | **55.58*** | **85.26*** |

inance on rigid tasks, it maintains competitive parity with explicit baselines, effectively bypassing the typical trade-off between creativity and accuracy.

**Overall superiority.** Table 3 reports the non-tie win rate ($\mathcal{R}^*_{win}$) against the strong explicit baseline (Stage 1+2+4). This primary comparison isolates the reasoning interface under a shared Stage-4 optimization procedure: GRPO provides the training method, while latent thinking or CoT define the interface used for reasoning. LatentChem demonstrates robust superiority, dominating on ChemCoTBench (**59.88%**) and ChEBI-20 (**85.26%**), while generalizing well to ChemLLMBench (**55.58%**). On Mol-Instructions, it maintains competitive parity (**49.88%**). These results confirm that the latent reasoning paradigm offers a superior substrate for complex chemical logic, enhancing capabilities without compromising foundational proficiency on standard tasks.

**Inference efficiency.** Beyond capability, we quantify reasoning step overhead by contrasting the total reasoning overhead (baseline CoT tokens vs. LatentChem's combined latent and textual steps). Here, one step means generating one latent token or one text token. As shown in Figure 8, by compressing verbose linguistic thought, LatentChem reduces reasoning step overhead by factors ranging from **5.4×** to **29.9×** (average **10.84×**, with a measured **5.96×** wall-

clock speedup; see Appendix K) for protocol and details results of the wall-clock latency experiment. This confirms that the latent paradigm is not only effectively superior but also computationally transformative, reducing reasoning overhead beyond traditional CoT.

We attribute this dramatic acceleration to the inherent compressibility of chemical logic in continuous space compared to language. Consider a routine optimization step, such as appending a functional group. In the latent manifold, the model can theoretically execute this structural shift via a single or few vector transitions, effectively "gliding" across the smooth landscape visualized in Figure 1 (b). Conversely, explicit CoT is forced to discretize this operation into a verbose textual chain (e.g., analyzing binding sites, verifying valency, and describing the addition), triggering dozens of redundant autoregressive forward passes to achieve the exact same chemical move (the jagged "staircase" trajectory in Figure 1 (c)). By bypassing this language bottleneck, LatentChem aligns computational expenditure with the actual complexity of the chemical task. Further analysis is presented in Appendix A.

### 5.3. Ablation Study

To isolate the drivers of performance, we conduct a component-wise ablation (Table 5). These ablations are trained from scratch. Removing Latent Thinking causes a

*Table 4.* Scale-matched evaluation against explicit-CoT baselines. Values are non-tie win rates (%) computed with the same metric as Table 3.

| Model | ChemCoTBench | | | Mol-Instructions | | | ChemLLMBench | | | ChEBI-20 |
|---|---|---|---|---|---|---|---|---|---|---|
| | Open | Closed | All | Open | Closed | All | Open | Closed | All | Open |
| LatentChem-4B | 55.41 | 41.28 | 52.42 | 48.37 | 52.18 | 51.93 | 64.29 | 56.38 | 57.03 | 57.07 |
| LatentChem-14B | 52.29 | 49.72 | 51.80 | 55.65 | 51.49 | 51.81 | 51.02 | 55.02 | 54.69 | 55.43 |

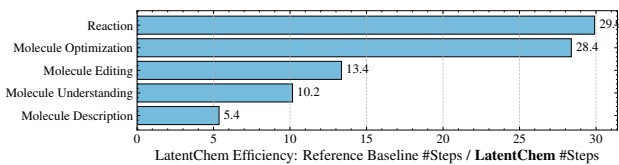

*Figure 8.* Inference efficiency analysis. We report the reasoning step overhead ratio (Explicit CoT Steps / LatentChem Steps) across all five task categories. LatentChem achieves up to **29.9×** reduction in reasoning step overhead on reaction tasks and a **10.84×** average reduction in reasoning step overhead (**5.96×** wall-clock speedup).

*Table 5.* Ablation study. We report the success rate (SR%) for molecule optimization and METEOR scores for molecule description across all applicable tasks from the four evaluated benchmarks.

| Method | Mol. Optimization (SR%↑) | Mol. Description (METEOR↑) |
|---|---|---|
| w/o Latent Thinking | 71.00 | 0.052 |
| w/o Latent Projector | 69.83 | 0.087 |
| w/o ChemUpdater | 68.67 | 0.068 |
| LatentChem (Full) | **80.67** | **0.143** |

precipitous performance drop, confirming that continuous vectors capture chemical nuances far more effectively than discrete tokens. Furthermore, the removal of ChemUpdater leads to a 12.0% decline in optimization success, underscoring that dynamic perception is essential for chemical reasoning. Finally, the latent projector proves indispensable for bridging the semantic gap; without it, the model fails to map internal thought vectors back to the input space, disrupting the reasoning trajectory.

### 5.4. Scale-Matched Model Evaluation

In this section, we further ask whether the advantage of the latent interface persists when the backbone scale changes. We therefore repeat the same non-tie win-rate comparison with matched 4B and 14B LatentChem models against explicit-CoT baselines of the same scale. As shown in Table 4, LatentChem remains favorable overall at both tested scales across the evaluated benchmarks.

### 6. Discussion and Conclusion

This study investigates the "modality mismatch" hypothesis in chemical LLMs, proposing that the discontinuity of natural language inhibits the modeling of continuous chemical manifolds. By introducing LatentChem, we demonstrate that decoupling reasoning from linguistic generation offers a superior computational substrate for chemical logic. Our analysis reveals the mechanism behind this success: the model exhibits spontaneous internalization, abandoning verbose textual derivations for compact latent computation that correlates with physical chemical topology. This structural advantage translates directly into performance, with LatentChem achieving a **59.88%** non-tie win rate against strong explicit baselines on ChemCoTBench and a **10.84×** reduction in reasoning step overhead (**5.96×** wall-clock speedup), effectively breaking the trade-off between reasoning depth and inference latency.

**Limitations and future directions.** The transition to latent reasoning introduces an inherent interpretability trade-off, as the internalized process sacrifices the transparency of explicit CoT. While our post-hoc analysis validates the physical grounding of these thought vectors, the reasoning process remains opaque to human users. This opacity can complicate scientific auditability and trust, especially when intermediate reasoning must be inspected for verification. We also observe task-dependent gains and regressions, and acknowledge broader source-level overlap through shared public databases, the lack of separate RL prompt data, underexplored sensitivity to reward-design choices across heterogeneous tasks, and GRPO training stability as limitations. Future work should focus on hybrid cognitive architectures (Kahneman, 2011): systems that leverage efficient latent thinking for heavy structural computation (System 1) while retaining the ability to decode these thoughts into explicit language when justification is required (System 2). Ultimately, within the chemistry-specific empirical scope of this paper, this work positions latent thinking not merely as an acceleration trick, but as a promising reasoning modality for future scientific AI systems.

## Acknowledgements

We thank the anonymous reviewers for their constructive feedback.

## Impact Statement

This work advances machine learning methods for scientific reasoning, with a focus on chemical reasoning and molecular design. By improving the efficiency and accuracy of chemical LLMs, LatentChem may support beneficial applications such as molecular property optimization, and computational assistance for drug discovery and materials science. The same capabilities may also introduce dual-use risks if molecule generation or optimization is applied to harmful compound design, unsafe synthesis planning, or unsupported decision-making in high-stakes scientific settings. Generated molecules should therefore be treated as research hypotheses that require domain-expert review and appropriate safety checks before any real-world use.

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

# A. Efficiency Analysis

To analyze the reasoning-step overhead reductions reported in the main text, we formalize reasoning as a trajectory discretization problem. Crucially, curvature-limited discretization arises in the *representation space* of a modality (where discrete steps are taken). This appendix provides an interpretive analysis under simplifying assumptions, rather than a formal proof of speedups or a quantitative predictive theory of runtime efficiency.

## A.1. Preliminaries and Assumptions

**Definition A.1** (Chemical Manifold). Let $\mathcal{M} = (\mathcal{S}, g)$ be a smooth $d$-dimensional Riemannian manifold representing the chemical state space, where $g$ encodes physicochemical properties.

**Definition A.2** (Optimal Reasoning Path). The logical derivation from a problem state $x_{\text{start}}$ to a solution $x_{\text{end}}$ is modeled as a unit-speed geodesic $\gamma : [0, L] \to \mathcal{M}$, parameterized by manifold arc length $s$, such that $\nabla_{\dot{\gamma}}\dot{\gamma} = 0$ and $\|\dot{\gamma}(s)\|_g = 1$.

**Assumption A.3** (Perfect Manifold Learning). We assume an idealized scenario where both the Explicit CoT model and the LatentChem model attempt to approximate the same optimal path $\gamma$. This isolates modality-dependent efficiency limits from model capacity and optimization.

**Definition A.4** (Modality Representation Trajectory). Each reasoning modality $\text{mod}$ induces a differentiable representation map $\Phi_{\text{mod}} : \mathcal{M} \to \mathbb{R}^{m_{\text{mod}}}$. Along the optimal path, the modality operates on the represented trajectory

$$c_{\text{mod}}(s) := \Phi_{\text{mod}}(\gamma(s)) \subset \mathbb{R}^{m_{\text{mod}}}.$$

**Definition A.5** (Valid Discrete Approximation). A discrete reasoning chain is specified by a partition $0 = s_0 < s_1 < \cdots < s_N = L$ and induces a polygonal approximation of $c_{\text{mod}}$ by connecting consecutive endpoints $c_{\text{mod}}(s_k)$ and $c_{\text{mod}}(s_{k+1})$ with a line segment in $\mathbb{R}^{m_{\text{mod}}}$. The approximation is *valid* if for each segment

$$\sup_{s \in [s_k, s_{k+1}]} \text{dist}\Big(c_{\text{mod}}(s), \overline{c_{\text{mod}}(s_k)c_{\text{mod}}(s_{k+1})}\Big) \leq \delta,$$

where $\text{dist}$ is Euclidean distance and $\delta > 0$ is a fixed tolerance.

**Definition A.6** (Effective Curvature in Representation Space). Let $\tilde{s}$ denote arc-length along the represented curve $c_{\text{mod}}$ in $\mathbb{R}^{m_{\text{mod}}}$. Assuming $c_{\text{mod}}$ is $C^2$, define curvature along the path by

$$\kappa_{\text{mod}}(s) := \left\| \frac{d^2 c_{\text{mod}}}{d\tilde{s}^2}(s) \right\|.$$

**Definition A.7** (Intrinsic Semantic Resolution (Path-Advance)). The intrinsic semantic resolution of modality $\text{mod}$ is the maximum *advance along the optimal path* that a single discrete step can make:

$$s_{k+1} - s_k \leq \rho_{\text{mod}} \qquad \text{for all } k. \tag{1}$$

Equivalently, one step can traverse at most $\rho_{\text{mod}}$ units of manifold arc-length along $\gamma$.

**Assumption A.8** (Non-degenerate Representation Speed). Along the optimal path, the representation does not collapse locally: there exists $v_{\min} > 0$ such that

$$\|c'_{\text{mod}}(s)\| \geq v_{\min} \qquad \text{for all } s \in [0, L]. \tag{2}$$

## A.2. The Curvature–Resolution Theorem

**Theorem A.9** (Curvature–Resolution Trade-off (Qualitative Lower Bound)). *Let $\rho_{\text{mod}}$ be the intrinsic semantic resolution (Definition A.7), and let $\kappa_{\text{mod}}(s)$ be the effective curvature of $c_{\text{mod}}(s) = \Phi_{\text{mod}}(\gamma(s))$ in representation space. Under Assumption A.8, any valid discrete approximation requires at least*

$$N \geq \int_0^L \max\left( \frac{1}{\rho_{\text{mod}}}, v_{\min}\sqrt{\frac{\kappa_{\text{mod}}(s)}{8\delta}} \right) ds. \tag{3}$$

*Consequently, the reasoning-step overhead reduction ratio between LatentChem and Explicit CoT can be estimated by*

$$\eta = \frac{N_{txt}}{N_{lat}} \approx \frac{\int_0^L \max\left( \frac{1}{\rho_{txt}}, v_{\min}\sqrt{\frac{\kappa_{txt}(s)}{8\delta}} \right) ds}{\int_0^L \max\left( \frac{1}{\rho_{lat}}, v_{\min}\sqrt{\frac{\kappa_{lat}(s)}{8\delta}} \right) ds}. \tag{4}$$

### A.3. Proof of Theorem A.9

**Step 1: Geometric constraint in representation arc length.** Fix an interval $[s_k, s_{k+1}]$ and let its manifold-arc-length width be $\Delta s_k := s_{k+1} - s_k$. Let the representation-space arc length of the curve segment be

$$\tilde{\ell}_k := \int_{s_k}^{s_{k+1}} \|c'_{\mathrm{mod}}(u)\| \, du.$$

Define $\kappa_k := \sup_{u \in [s_k, s_{k+1}]} \kappa_{\mathrm{mod}}(u)$. A standard chord–arc deviation (sagitta) bound for $C^2$ curves, stated in terms of representation arc length, gives

$$\sup_{s \in [s_k, s_{k+1}]} \mathrm{dist}\Big(c_{\mathrm{mod}}(s), \overline{c_{\mathrm{mod}}(s_k)c_{\mathrm{mod}}(s_{k+1})}\Big) \leq \frac{1}{8} \kappa_k \tilde{\ell}_k^2. \tag{5}$$

Validity implies $\tilde{\ell}_k \leq \sqrt{\frac{8\delta}{\kappa_k}}$.

**Step 2: Converting to a bound on manifold-arc-length advance.** By Assumption A.8, $\tilde{\ell}_k \geq v_{\min}\Delta s_k$, hence

$$\Delta s_k \leq \frac{1}{v_{\min}} \sqrt{\frac{8\delta}{\kappa_k}}. \tag{6}$$

Independently, Definition A.7 gives the modality advance constraint

$$\Delta s_k \leq \rho_{\mathrm{mod}}. \tag{7}$$

Combining (6)–(7) yields

$$\Delta s_k \leq \min\left(\rho_{\mathrm{mod}}, \frac{1}{v_{\min}} \sqrt{\frac{8\delta}{\kappa_k}}\right).$$

**Step 3: Aggregation into a global lower bound.** Define

$$w(s) := \max\left(\frac{1}{\rho_{\mathrm{mod}}}, v_{\min}\sqrt{\frac{\kappa_{\mathrm{mod}}(s)}{8\delta}}\right).$$

On each interval $[s_k, s_{k+1}]$, we have $\kappa_{\mathrm{mod}}(s) \leq \kappa_k$, so

$$w(s) \leq \max\left(\frac{1}{\rho_{\mathrm{mod}}}, v_{\min}\sqrt{\frac{\kappa_k}{8\delta}}\right) \quad \text{for all } s \in [s_k, s_{k+1}].$$

Therefore,

$$\int_{s_k}^{s_{k+1}} w(s)\, ds \leq \Delta s_k \cdot \max\left(\frac{1}{\rho_{\mathrm{mod}}}, v_{\min}\sqrt{\frac{\kappa_k}{8\delta}}\right) \leq 1,$$

and summing over $k$ yields $\int_0^L w(s)\, ds \leq N$, proving (3). $\qquad\square$

### A.4. Efficiency Scaling via Manifold Rectification

Theorem A.9 implies a qualitative trade-off: to keep chord deviation within tolerance $\delta$, high representation-space curvature forces smaller advances $\Delta s_k$ (hence more steps), while a rectified representation reduces curvature and permits coarser discretization.

**Latent Space ($\kappa_{lat} \to 0$).** If representation learning rectifies the latent trajectory so that $\kappa_{lat}(s) \approx 0$, then the curvature term vanishes and

$$N_{lat} \gtrsim \int_0^L \frac{1}{\rho_{lat}}\, ds = \frac{L}{\rho_{lat}}.$$

**Explicit CoT Space (curvature remains large).** If the text-space representation retains high curvature $\kappa_{txt}(s)$, then for sufficiently complex tasks the curvature term can dominate, leading to

$$N_{txt} \gtrsim \frac{1}{v_{\min,txt}} \int_0^L \sqrt{\frac{\kappa_{txt}(s)}{8\delta}} \, ds,$$

and hence a larger step count than the rectified latent case. This supports the qualitative hypothesis that LatentChem's advantage grows with task complexity when it reduces representation-space curvature.

# B. Benchmark Details and Task Definition

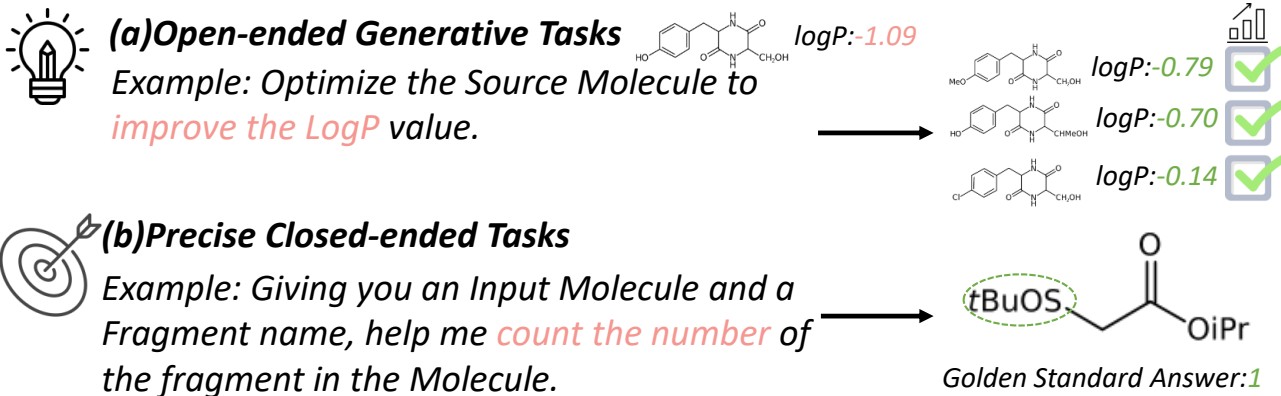

*Figure 9.* Taxonomy of chemical reasoning tasks. The evaluated tasks consist of two categories: (a) open-ended generative tasks (*e.g.,* molecule optimization), characterized by the non-determinism of the optimal solution; (b) precise closed-ended tasks (*e.g.,* fragment counting), which demand a unique, deterministic standard answer.

## B.1. ChemCoTBench

ChemCoTBench is a step-by-step, application-oriented, and high-quality benchmark for assessing LLM reasoning in chemical applications. ChemCoTBench comprises molecule understanding, molecule editing, molecule optimization, and reaction prediction, covering 22 subtasks and 1,495 examples. These samples are derived from authoritative public databases (*e.g.,* PubChem, ChEMBL, ZINC) and patent corpora, carefully reviewed by the combined effort of LLM and 13 chemists for chemical validity and diversity.

**Molecule understanding.** As shown in Figure 10(a), we evaluate models on progressively more demanding levels of structural comprehension. First, models must recognise and count two fundamental molecular features: (i) functional groups (FGs), which are atom clusters that largely determine a molecule's physicochemical properties and reactivity; and (ii) rings, which impose fixed conformations and serve as common building blocks in drug design, crystal engineering, and polymer synthesis. Accurate identification and counting of functional groups and rings requires both lexical and syntactic understanding of SMILES and remains challenging for LLMs that lack explicit topological awareness. Second, we assess recognition of higher-order scaffolds: Murcko scaffolds—obtained by systematically removing side chains and used for structural analysis in medicinal chemistry—and ring systems, including fused and bridged motifs that present substantial synthetic challenges. These tasks probe hierarchical and topology-aware representations. Finally, we introduce SMILES-equivalence tasks (permutations and mutations) to test whether models can identify chemically equivalent structures despite surface-level SMILES variability, thereby evaluating robustness to representational changes.

**Molecule editing.** This suite (Figure 10(b)) measures whether models can execute basic molecular-editing operations (add, delete, substitute functional groups) when guided by natural-language instructions. These operations are analogous to elementary arithmetic in mathematics and form the primitive actions required for more complex design workflows. Many optimization and synthesis problems can be decomposed into sequences of such edits; thus this task evaluates two core capabilities: (i) preserving chemical validity after each edit, and (ii) correctly applying the requested modification.

**Molecule optimization.** Here we test whether models can propose structural modifications that improve a specified target (Figure 10(c)). We consider two classes of objectives. At the physicochemical level, targets include LogP, solubility, and QED to assess improvements in drug-likeness. At the target-specific level, we consider binding-affinity–driven objectives for receptors such as DRD2, GSK3$\beta$, and JNK3, which require reasoning about drug–target interactions. Success on these tasks requires not only syntactic parsing of molecular structure but also inference about how local and global modifications affect complex chemical and biological properties.

**Molecule description.** This task (Figure 10(d)) evaluates whether models can generate accurate and informative natural-language descriptions of molecular structures. Given a molecular representation (*e.g.,* SMILES or graph-based encoding),

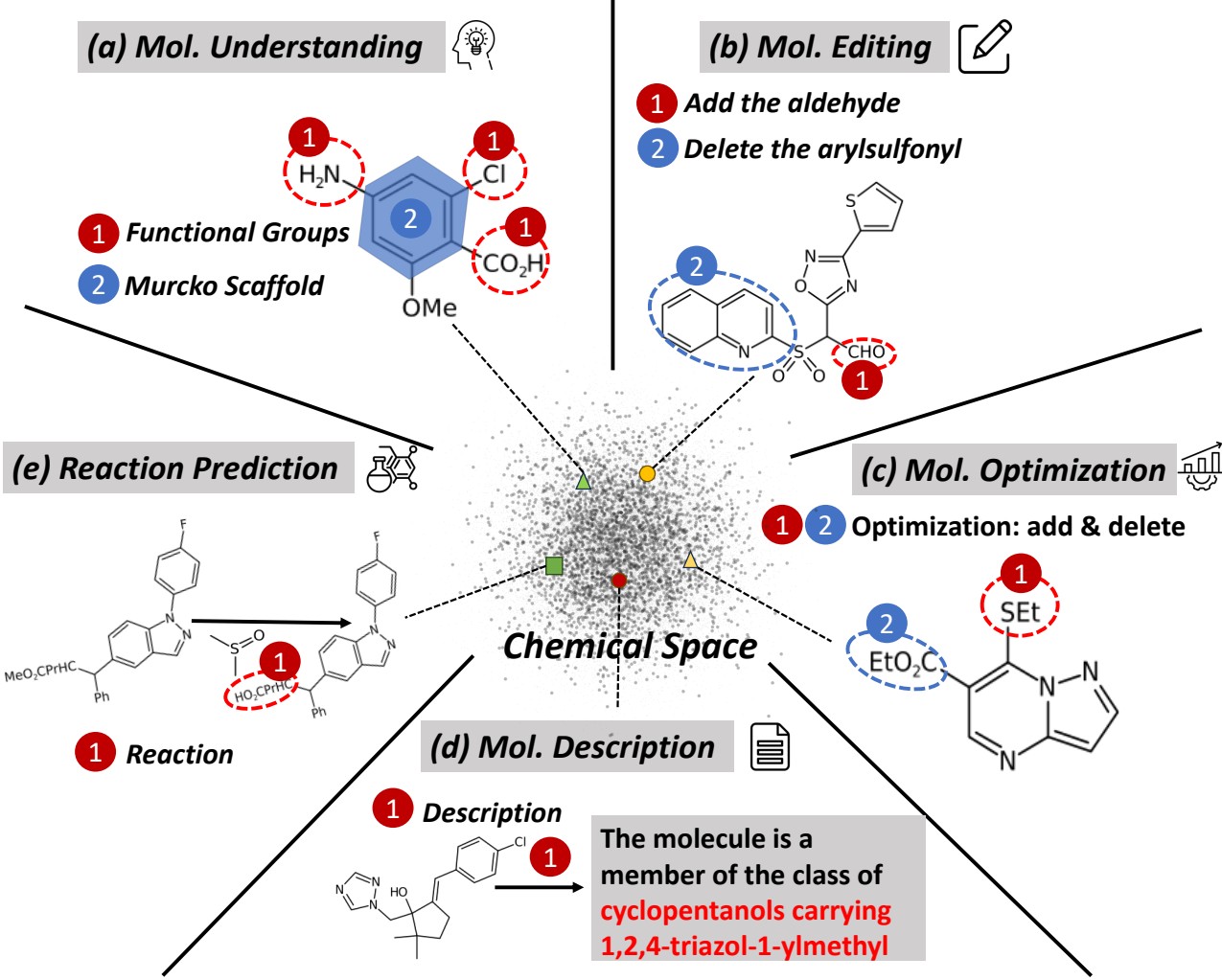

*Figure 10.* Task overview of ChemCoTBench

models are required to describe key chemical features such as functional groups, ring systems, heteroatom composition, stereochemistry, and overall structural motifs, and in some cases relate these features to physicochemical or biological properties. Successful performance demands precise parsing of molecular structure, abstraction of salient substructures, and coherent translation of symbolic chemical information into human-readable descriptions, reflecting a deep alignment between molecular representation and chemical semantics.

**Reaction prediction.** As shown in Figure 10(e), this set of tasks evaluates chemical reasoning across multiple granularities: (i) *Forward prediction*: given reactants and reagents, predict the major products and plausible by-products, reflecting knowledge of reactivity, selectivity, and thermodynamic/kinetic considerations; (ii) *Single-step retrosynthesis*: propose reactants and disconnections for a target product, identifying key bond cleavages and feasible transformations; (iii) *Reaction-condition recommendation*: suggest catalysts, solvents, and reagents appropriate for a specified transformation, accounting for solvent effects and catalyst mechanisms that influence yield and selectivity; and (iv) *Mechanistic understanding*: include next-elementary-step prediction (intermediate species and their formation) and mechanism-route selection (choose the most plausible pathway among alternatives). Together, these tasks span coarse product-level prediction to fine-grained mechanistic reasoning, providing a comprehensive benchmark of an LLM's ability to act as a chemical-reasoning agent.

## B.2. Mol-Instructions

Mol-Instructions is a large instruction dataset for biomolecular tasks. It is organized into three components—molecule-oriented , protein-oriented, and biomolecular-text; data were derived from various licensed biochemical resources and generated via template conversion and human–AI augmentation. Among all of the three components, only the molecule-oriented subset is related to our work.

**Molecular description generation.** Molecular description generation evaluates the model's high-level comprehension of molecules. Given a molecule's SMILES string, the model is required to infer the molecule's structure, properties, biological activities, and potential applications. This task is evaluated using standard natural-language metrics (*e.g.,* BLEU, ROUGE, METEOR).

**Description-guided molecule generation.** This task requires the model to design new molecules from natural-language descriptions. Given descriptions of desired structural features, properties, biological activity, or applications, the model should generate the corresponding molecular SMILES representation.

**Property prediction tasks.** Property prediction refers to the inference or estimation of a molecule's intrinsic physicochemical properties using information derived from its structural features. This task supports the high-throughput assessment of a wide range of molecular properties, thereby enabling efficient virtual screening of compound libraries. Moreover, it allows for the prediction of previously unknown properties of novel molecules, enhancing research productivity and substantially shortening development timelines.

**Reaction prediction tasks.** Reaction prediction tasks comprise three independent problems: forward reaction prediction, retrosynthesis, and reagent prediction. In *forward reaction prediction*, the model is given reactant and reagent SMILES and must predict the major product SMILES. In *retrosynthesis*, the model is given a product SMILES and must predict plausible reactant SMILES. In *reagent prediction*, the model is given both reactant and product SMILES and must propose suitable catalysts, solvents, or ancillary substances required to effect the specified chemical reaction.

## B.3. ChEBI-20

ChEBI-20 was constructed by extracting PubChem compound records annotated with ChEBI and pairing each compound with its ChEBI textual description. To increase informational content and exclude trivial entries, only records whose descriptions exceed 20 tokens were retained. Compounds whose SMILES could not be parsed by RDKit were removed to ensure consistent molecular representations. The final curated dataset contains 33,010 molecule–text pairs and is referred to as ChEBI-20, which has been commonly used as a benchmark for text–molecule retrieval, molecule captioning, and text-guided molecular generation.

**Molecule caption generation.** We evaluate models on *molecule caption generation* to assess molecular understanding. Given a molecular SMILES string, the model must generate a detailed natural-language description that accurately reports salient structural features, functional groups, and chemically relevant properties present in the input. Evaluation combines standard natural-language quality metrics (BLEU, ROUGE, METEOR).

## B.4. ChemLLMBench

ChemLLMBench is a comprehensive benchmark that evaluates large language models on eight practical chemistry tasks—including name prediction, molecular property prediction, yield and reaction prediction, retrosynthesis, text-based molecule design, molecule captioning, and reagent selection—using widely recognized datasets (*e.g.,* MoleculeNet, USPTO, PubChem), assessing different abilities such as understanding, reasoning, and explaining.

**Molecule captioning.** Molecule captioning evaluates the model's ability to explain molecular structures. Given a molecule's SMILES string, the model is required to generate a textual description of its key features, properties, biological activities, and potential applications. This task is evaluated using standard natural-language metrics (*e.g.,* BLEU, ROUGE, METEOR).

**Text-based molecule design.** This task requires the model to design new molecules from natural-language descriptions. Given textual descriptions of desired properties, structural features, biological activity, or applications, the model should generate the corresponding molecular SMILES representation.

**Reaction-related tasks.** Reaction-related tasks comprise four independent problems: yield prediction, forward reaction prediction, retrosynthesis, and reagents selection. In *yield prediction*, the model estimates if a reaction is high-yielding given reactants and reagents. In *forward reaction prediction*, the model is given reactant and reagent SMILES and must predict the major product SMILES. In *retrosynthesis*, the model is given a product SMILES and must predict plausible reactant SMILES. In *reagents selection*, the model selects appropriate reagents, ligands, or solvents from candidates for a given reaction.

# C. Detailed Training Protocol

**Stage 1: Establishing the Molecular-Linguistic Mapping**   The primary goal of this stage is to align the projected molecular representations $\mathbf{H}_{chem}$ with the semantic space of the pre-trained LLM. To establish a robust mapping, we utilize instruction-tuning data containing reasoning chains but intentionally *suppress* the intermediate reasoning steps during this phase. Specifically, given a sample comprising an instruction $\mathbf{x}_{prompt}$, a chain-of-thought (CoT), and a final answer $\mathbf{y}_{ans}$, we train the model to directly generate the final answer. This "answer-only" supervision acts as an information bottleneck, compelling the projector to compress all necessary chemical properties into the ChemTokens.

To ensure the model genuinely utilizes these molecular features rather than memorizing textual patterns, we employ a **Counterfactual Alignment** strategy. Let $\mathcal{L}_{CE}(\mathbf{y}|\mathbf{H})$ denote the standard cross-entropy loss. We define the loss on the original clean input as $\mathcal{L}_{clean} = \mathcal{L}_{CE}(\mathbf{y}_{ans}|\mathbf{H}_{chem}, \mathbf{H}_{prompt})$. Simultaneously, we perform a corrupted forward pass using perturbed ChemTokens $\tilde{\mathbf{H}}_{chem} = \Phi(\mathbf{H}_{chem})$ (via dropout, shuffling, or noise) to compute $\mathcal{L}_{corrupt} = \mathcal{L}_{CE}(\mathbf{y}_{ans}|\tilde{\mathbf{H}}_{chem}, \mathbf{H}_{prompt})$. The counterfactual alignment loss is defined as a hinge loss:

$$\mathcal{L}_{CF} = \max(0, \gamma - (\mathcal{L}_{corrupt} - \mathcal{L}_{clean})) \tag{8}$$

where $\gamma$ is a margin hyperparameter. The total objective for Stage 1 is:

$$\mathcal{L}_{total}^{(1)} = \mathcal{L}_{clean} + \lambda \mathcal{L}_{CF} \tag{9}$$

**Stage 2: SFT for Molecule-aware CoT**   In this stage, we transition from direct mapping to complex reasoning. We unlock the full potential of the data by training the model on the complete sequences, requiring it to generate the explicit CoT derivations prior to the final answer. This enables the model to decompose complex molecular problems into intermediate steps.

To prevent "hallucinated reasoning" and ensure that the generated logic remains grounded in the molecular structure, we continue to employ the counterfactual strategy. The target sequence is extended to include both the reasoning chain and the answer, denoted as $\mathbf{y}_{full} = [\mathbf{y}_{cot}, \mathbf{y}_{ans}]$. Accordingly, the clean and corrupted losses are updated to $\mathcal{L}_{clean} = \mathcal{L}_{CE}(\mathbf{y}_{full}|\mathbf{H}_{chem}, \mathbf{H}_{prompt})$ and $\mathcal{L}_{corrupt} = \mathcal{L}_{CE}(\mathbf{y}_{full}|\tilde{\mathbf{H}}_{chem}, \mathbf{H}_{prompt})$. The total objective for Stage 2 becomes:

$$\mathcal{L}_{total}^{(2)} = \mathcal{L}_{clean} + \lambda \mathcal{L}_{CF} \tag{10}$$

This formulation ensures that the model not only generates the correct answer but also derives it through a valid, molecule-dependent reasoning path.

**Stage 3: Chemistry-aware Latent Mind Activation**   In this phase, we activate the latent thinking modules during training. A major challenge in training these lightweight modules alongside a massive LLM backbone is the risk of optimization imbalance: the LLM's overwhelming parameter count can easily lead it to bypass the latent signals, treating them as noise rather than meaningful thought vectors. To mitigate this, we employ a **Strict Freezing Strategy**. We freeze both the Chemical Adapter and the LLM backbone, updating *only* the parameters of the thinker and updater modules. This constraint forces the latent modules to adapt to the frozen semantic space of the LLM, compelling them to generate "legible" latent thoughts that effectively guide the LLM's frozen decoder toward the correct reasoning path defined by the CoT data.

**Stage 4: GRPO with Latent Thinking Budget**   With the latent mind activated, we employ GRPO to refine the model's policy. Reversing the freezing strategy from the previous stage, we now **freeze** the latent thinking modules and fine-tune the remaining parameters. This strategy treats the learned latent dynamics as a stable "internal simulator," optimizing the backbone LLM to effectively utilize these latent thoughts for decision-making. The optimization is guided by a composite reward function comprising four terms: format adherence, answer validity, and answer correctness.

# D. Experimental Implementation Details

### D.1. Dataset and Preprocessing

**Prompt and Label Formatting.** The dataloader structures each training example to enforce specific output formats. Every question is appended with the instruction: "*Your final answer must be formatted as* `<answer> ... </answer>`". Ground-truth labels are formatted similarly. The prompt and response are tokenized separately and concatenated.

### D.2. Model Architecture

**Base Language Model.** We use Qwen-3-8B as the backbone LLM.

**Molecule Encoder.** We utilize SMI-TED Light as the molecule encoder. The encoder is kept frozen during all training stages and operates in inference mode (no gradient computation).

**Chemical Adapter.** We employ a Querying Transformer (Q-Former) style projector with 128 learnable queries, an input dimension of 768, and 8 attention heads. For each input molecule, the adapter produces a sequence of 128 projected tokens, encapsulated by special tokens `<mol_start>` and `<mol_end>`.

**Latent Modules.** We add specific control tokens to the vocabulary, including `<latent>`, `<start_latent>`, and `<end_latent>`.

### D.3. Supervised Fine-Tuning (SFT)

We adopt a three-stage SFT pipeline. All stages are trained using distributed data parallel (DDP) across 8 H200 GPUs using `bfloat16` precision.

**Common Training Configuration.** We use the AdamW (8-bit) optimizer with a cosine learning rate scheduler, a weight decay of 0.01, and gradient checkpointing enabled. The maximum sequence length is set to 8192 tokens. The regularization weight $\lambda = 0.2$, and $m = 0.1$ is the margin.

**Stage 1.** We train the LoRA adapters on the Qwen backbone and the Chemical Adapter for 3 epochs with a learning rate of $2 \times 10^{-4}$ and a batch size of 4 per device.

**Stage 2.** Weights are initialized from the Stage I checkpoint. We continue training the LoRA adapters and the Chemical Adapter for 3 epochs with a learning rate of $2 \times 10^{-4}$.

**Stage 3.** This stage incorporates the continuous reasoning loop. We insert a block of latent tokens between the prompt and the response: `<start_latent>` $+ N \times$ `<latent>` $+$ `<end_latent>`. The number of latent tokens $N$ is dynamic, calculated based on the length of the ground-truth CoT (capped at 4 tokens). Training proceeds for 3 epochs with a learning rate of $2 \times 10^{-4}$. The latent tokens are masked in the loss, except for the boundary token.

**Coconut-Chem Baseline.** Coconut-Chem uses the same training data and hyperparameters as LatentChem. It is first trained with our Stage 1 and Stage 2 protocol, and then trained with Coconut's official latent-reasoning procedure.

### D.4. Reinforcement Learning via GRPO (Stage 4)

In the final stage, we employ Group Relative Policy Optimization (GRPO) to further refine the model's reasoning capabilities.

**Algorithm and Environment.** We adapted the TRL GRPO implementation with vLLM for efficient rollout generation (tensor parallel size 1, GPU memory utilization 0.3). We sample a group size of $G = 8$ completions per prompt. Training runs for 1 epoch with a learning rate of $1 \times 10^{-5}$ and an effective batch size of 128 prompts per step. We use a KL penalty of $\beta = 0.0$ and a clipping epsilon of 0.2.

**Reward Functions.** We utilize a composite reward signal consisting of three equally weighted components:

1. **Format Reward:** +1.0 if the completion contains non-empty `<answer>` tags.

2. **Type Validity Reward:** +0.5 if the content within the tags matches the expected data type (e.g., valid SMILES string, float, or boolean).

3. **Correctness Reward:** +2.0 if the answer matches the ground truth. Correctness is determined via task-specific oracles, including exact canonical SMILES matching for structure generation and functional group verification for editing tasks.

**Generation Settings.** During rollouts, we use a temperature of 1.5 and top-p sampling of 0.9, with a maximum completion length of 2048 tokens.

# E. Metrics

**Mean Absolute Error:** The mean of the absolute deviations between predicted and true values, quantifies the typical magnitude of prediction error in regression tasks. Employed on *Functional Groups (FGs)* and *Rings*.

**Scaffold Similarity:** Structural conservation between generated and reference molecules is measured via the Tanimoto coefficient computed on molecular scaffolds. Scores range from 0 to 1, with higher values indicating stronger preservation of the core framework. Used in *Murcko scaffolds*.

**Accuracy:** The proportion of correctly predicted outcomes, serving as a baseline measure of overall correctness. For *Molecule Editing* tasks (e.g. add, delete, substitute functional groups), an outcome is correct only when the edited molecule stays valid, and the functional group count matches the instruction. Binary classification accuracy is calculated on *Ring System*. For reaction-prediction tasks (e.g. *Forward Prediction* and *Retrosynthesis*), we use Top-1 accuracy, i.e., a prediction is counted as correct only when the model's highest-ranked output exactly matches the ground-truth molecule(s).

**Improvement:** Absolute gains in the target property induced by *Molecule Optimization*. We mainly report the mean improvement across samples to indicate typical uplift.

**success rate:** The fraction of optimized molecules that exceed a predefined target threshold (for example, solubility over 0.8), reflecting the practical utility in *Molecule Optimization* tasks.

**Fingerprint-based Similarities (FTS):** Morgan (Schneider et al., 2015), MACCS (Durant et al., 2002), and RDKit fingerprints are used to reflect correctness and structural similarity between the predicted molecule and the reference molecule in reaction-prediction tasks (e.g. *Forward Prediction* and Retrosynthesis). Additionally, the FTS metric refers to the average of Morgan, MACCS and RDKit fingerprint-based similarities.

**Validity:** The proportion of generated SMILES strings that are syntactically valid and can be parsed into chemical structures (*e.g.,* successfully converted to RDKit molecule objects), indicating the chemical feasibility of model outputs. Reported explicitly in reaction-prediction tasks (e.g. *Forward Prediction* and Retrosynthesis).

**Text Similarities:** Text similarity metrics like BLEU (Papineni et al., 2002), ROUGE (Lin, 2004) and METEOR (Banerjee & Lavie, 2005) are utilized to assess the quality of generated molecule descriptions in *Molecular Description Generation* tasks by comparing them with the reference answers.

# F. Results for Closed-ended Tasks

We report the results for closed-ended tasks on ChemCoTBench, Mol-Instructions and ChemLLMBench to complement the results discussed in Section 5.2. Specifically, results on ChemCoTBench, including Molecule Understanding, Molecule Editing, and Reaction-related tasks are presented in Table 6. In addition, results on Mol-Instructions and ChemLLMBench, which comprise broad reaction-related subtasks, are shown in Table 7.

*Table 6.* Detailed performance on closed-ended tasks from ChemCoTBench. Metrics include mean absolute error (MAE↓) for functional group, Scaffold Similarity for Murcko scaffolds, correct rate (CR%) for ring system count, alongside Top-1 accuracy and fingerprint-based similarities (FTS) for reaction.

| Category | Method | Molecule Understanding | | | | Molecule Editing | | | Reaction | | | |
| | | Functional Group | | Scaffold | | Add | Delete | Sub | Fwd $_{major}$ | | Fwd $_{by}$ | |
| | | FG↓ | Ring↓ | Murcko↑ | Ring-sys↑ | CR% | CR% | CR% | Top1↑ | FTS↑ | Top1↑ | FTS↑ |
|---|---|---|---|---|---|---|---|---|---|---|---|---|
| Text-only | Qwen-3-8B | 7.03 | 1.03 | 0.02 | 36 | 4 | 52 | 1 | 0.00 | 0.02 | 0.00 | 0.00 |
| | Qwen-3-8B (SFT) | 0.12 | 0.45 | 0.40 | 65 | 55 | 76 | 37 | 0.09 | 0.35 | 0.02 | 0.07 |
| Chem. LLMs | Stage 1 | 0.14 | **0.10** | 0.71 | 83 | 62 | 61 | 47 | 0.15 | 0.57 | **0.12** | 0.18 |
| | Stage 1+2 | 0.08 | 0.20 | 0.66 | 63 | 63 | 80 | 49 | 0.10 | 0.44 | 0.03 | 0.11 |
| | Stage 1+2+4 | **0.07** | 0.15 | 0.65 | 83 | 70 | **88** | 60 | **0.20** | **0.63** | **0.12** | **0.20** |
| Latent Chem. LLMs | Coconut-Chem | 0.08 | 0.30 | 0.64 | 60 | 54 | 68 | 44 | 0.08 | 0.42 | 0.02 | 0.09 |
| | LatentChem (Ours) | 0.09 | 0.15 | **0.81** | **97** | **71** | 84 | **61** | 0.17 | 0.52 | 0.08 | 0.17 |

*Table 7.* Detailed performance on closed-ended tasks from Mol-Instructions and ChemLLMBench. Metrics include Top-1 accuracy and fingerprint-based similarities (FTS) for reaction prediction, alongside correct rate (CR%) for reagent selection.

| Category | Method | Mol-Instructions | | | | | | ChemLLMBench | | | | | | |
| | | Reagent | | Fwd$_{Major}$ | | Retro | | Reagent Sel. | | | Fwd $_{major}$ | | Retro | |
| | | Top1↑ | FTS↑ | Top1↑ | FTS↑ | Top1↑ | FTS↑ | Ligand% | Reactant% | Solvent% | Top1↑ | FTS↑ | Top1↑ | FTS↑ |
|---|---|---|---|---|---|---|---|---|---|---|---|---|---|---|
| Text-only | Qwen-3-8B | **0.00** | 0.02 | 0.00 | 0.04 | **0.00** | 0.05 | 9 | 11 | 14 | 0.00 | 0.02 | **0.00** | 0.07 |
| | Qwen-3-8B (SFT) | **0.00** | 0.06 | 0.02 | 0.26 | **0.00** | 0.30 | 31 | 43 | 23 | 0.02 | 0.29 | **0.00** | 0.31 |
| Chem. LLMs | Stage 1 | **0.00** | **0.07** | 0.05 | **0.55** | **0.00** | 0.47 | 28 | **51** | 34 | 0.01 | 0.49 | **0.00** | 0.47 |
| | Stage 1+2 | **0.00** | **0.07** | 0.05 | 0.42 | **0.00** | 0.42 | 34 | 37 | 31 | **0.04** | 0.38 | **0.00** | 0.46 |
| | Stage 1+2+4 | **0.00** | **0.07** | **0.09** | 0.52 | **0.00** | 0.49 | **47** | 39 | 33 | **0.04** | 0.48 | **0.00** | 0.45 |
| Latent Chem. LLMs | Coconut-Chem | **0.00** | **0.07** | 0.02 | 0.36 | **0.00** | 0.36 | 31 | 48 | 31 | 0.01 | 0.35 | **0.00** | 0.38 |
| | LatentChem (Ours) | **0.00** | **0.07** | 0.03 | 0.50 | **0.00** | **0.52** | 13 | 46 | **49** | 0.02 | **0.53** | **0.00** | **0.49** |

# G. Additional Results for Causal Study

This appendix contains a supplementary figure supporting the causal analysis reported in Section 4.2. Omitted from the main text for reasons of space, Figure 11 supplements Figure 5 by providing additional measurements of generated CoT length.

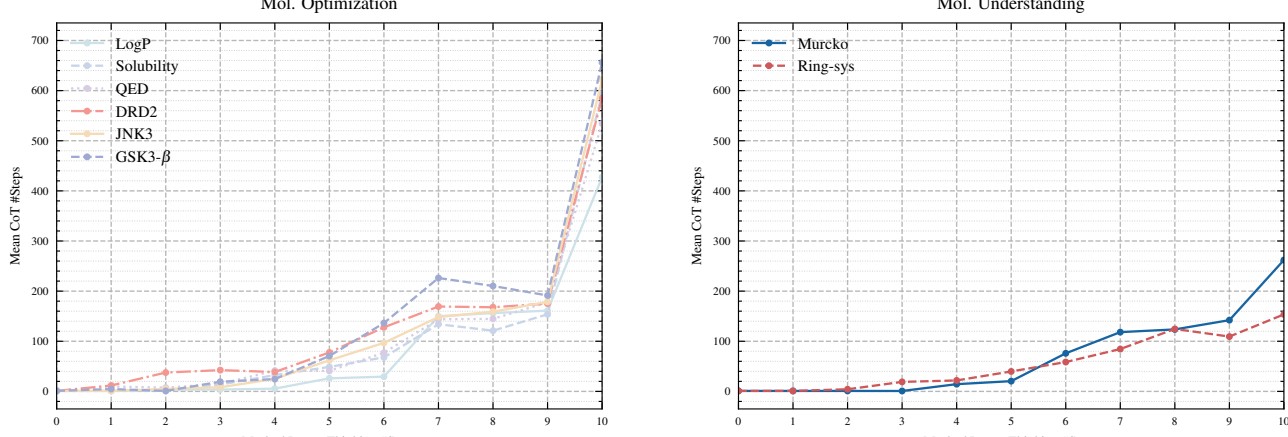

*Figure 11.* Additional results for causal necessity analysis. We monitor the length of generated explicit CoT (y-axis) as the first $k$ latent tokens are replaced with Gaussian noise.

## H. Additional Case Study on Latent Thinking Budget Test

To better demonstrate the spontaneous CoT internalization ability of our model discussed in Section 4.1, we present the full, unedited model outputs with different latent thinking budget for a representative case from the solubility optimization subtask alongside with the textual prompt, as illustrated in Figure 12.

---

**User:**

You are an expert chemist. Below is a molecule structure: `CCNc1ccc(S(=O)(=O)c2ccc(N)cc2)cc1`. Optimize the Source Molecule to improve the compound's ability to dissolve in water (Solubility) while following a structured intermediate optimization process. Your final answer must be formatted as `<answer> SMILES </answer>`.

- - - - - - - - - - - - - - - - - - - - - - - - - - - - - - - -

**Latent Thinking Budget 10:**

`.<answer> Nc1ccc(S(=O)(=O)c2ccc(N)cc2)cc1 </answer>`

**Latent Thinking Budget 0:**

**Step 1: Structural Analysis of Source Molecule:** Structural Analysis of Source Molecule: The source molecule contains a sulfonamide group (`S(=O)(=O)`) attached to a benzene ring, which is further substituted with a methyl-piperazine group (`CCN`). The sulfonamide is linked to another benzene ring with an amino (`-NH2`) substituent. The molecule has hydrophobic (methyl-piperazine and aromatic rings) and moderately polar (sulfonamide, amino group) regions.

**Step 2: Property Analysis:** The solubility of the molecule is limited by its hydrophobic methyl-piperazine group and the overall size of the aromatic systems. The sulfonamide and amino groups provide some water solubility but are insufficient for optimal aqueous dissolution.

**Step 3: Limitation in Source Molecule for Property:** The methyl-piperazine group (`CCN`) is a major contributor to low solubility due to its hydrophobic character and potential steric hindrance. The single amino group on the secondary benzene ring offers limited hydrogen bonding capacity.

**Step 4: Optimization for Source Molecule:** 1. Replace the methyl-piperazine group (`CCN`) with a smaller, polar cyclopropylamine group (`NCC1CC1`) to reduce hydrophobicity and steric bulk. 2. Maintain the sulfonamide and amino groups for polarity. 3. Simplify the substituent structure to enhance solvation.

**Step 5: Final Target Molecule:** `Nc1ccc(S(=O)(=O)c2ccc(NCC3CC3)cc2)cc1`

`<answer> Nc1ccc(S(=O)(=O)c2ccc(NCC3CC3)cc2)cc1 </answer>`

---

*Figure 12.* Comparison of model output with maximum latent thinking steps set to 10 versus 0.

# I. Additional Statistics on Reasoning-Step Overhead Reduction

In this appendix, Figure 13 serves as a per-benchmark per-task breakdown of the aggregated results on LatentChem's reasoning-step overhead reduction reported in Figure 8. Within each task category, the measured overhead reduction remains reasonably consistent across the evaluated benchmarks, supporting the use of averaged results in the main paper.

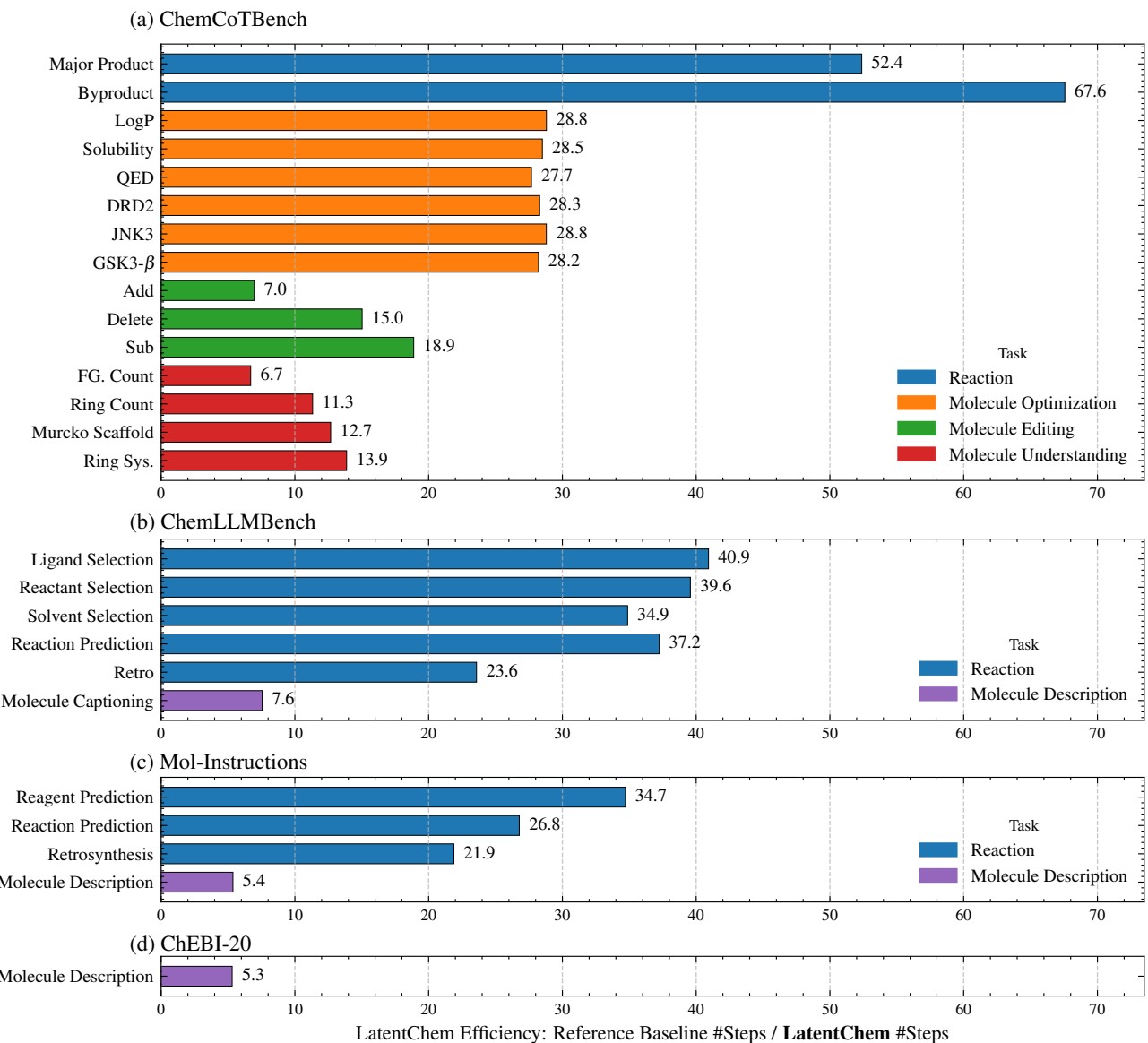

*Figure 13.* Detailed reasoning-step overhead reduction statistics grouped by benchmark. Task categories are sorted by decreasing *LatentChem reasoning-step overhead reduction*. Within each category, the task order remains consistent with that of other tables and figures in this study.

# J. Comparison with SOTA Models on ChemCoTBench

We adapt selected evaluation tables from ChemCoTBench (Hao et al., 2025a), limiting the reported sub-tasks to those evaluated in our experiments. Specifically, we omit the equivalence subtask from Molecule Understanding, as well as Retrosynthesis, RCR, NEPP, and MechSel from the Chemical Reaction task. Alongside the results reported in the original benchmark, we present the performance of our model on the same sub-tasks for reference. The results are shown in Tables 8, 9, and 10 respectively.

*Table 8.* Comparison with results reported on the ChemCoTBench foundational tasks, including molecule understanding, molecule editing, and their correlated subtasks. For the functional-group counting task (FG) and ring counting task (Ring) in the functional-group level molecule understanding, we apply the mean absolute error (MAE) as the evaluation metric. Tanimoto molecule similarity is applied as the evaluation for the Murcko scaffold extraction task (Murcko). The accuracy (%) metric is applied to other subtasks.

| Models | Func-Group | | Scaffold | | Molecule-Edit | | |
|---|---|---|---|---|---|---|---|
| | FG↓ | Ring↓ | Murcko↑ | Ring-sys↑ | Add | Delete | Sub |
| *W/ Thinking* | | | | | | | |
| Gemini-2.5-pro-think | 0.11 | 0.60 | 0.51 | 87.5 | 100 | 85 | 81.7 |
| Claude3.7-sonnet-think | 0.21 | 1.60 | 0.40 | 80.0 | 85 | 80 | 83.4 |
| DeepSeek-R1 | 0.27 | 1.55 | 0.34 | 45.0 | 70 | 70 | 68.3 |
| o3-mini@20250103 | 0.13 | 0.60 | 0.39 | 75.0 | 65 | 55 | 80.0 |
| o1-mini@20240912 | 0.21 | 1.25 | 0.25 | 61.7 | 55 | 80 | 58.3 |
| Qwen3-235B-A22B-think | 0.42 | 1.00 | 0.38 | 82.5 | 40 | 75 | 71.7 |
| Qwen3-32B-think | 0.25 | 0.95 | 0.21 | 75.0 | 20 | 55 | 20.0 |
| Llama-Nemo-49B-think | 0.80 | 1.90 | 0.09 | 86.8 | 0 | 80 | 8.0 |
| *W/o Thinking* | | | | | | | |
| GPT-4o@20241120 | 0.17 | 1.35 | 0.21 | 80.0 | 80 | 80 | 65.0 |
| Deepseek-V3 | 0.15 | 1.50 | 0.24 | 76.7 | 70 | 75 | 76.7 |
| Gemini-2.0-flash | 0.19 | 1.65 | 0.43 | 75.0 | 65 | 75 | 66.7 |
| Qwen3-235B-A22B | 0.42 | 1.00 | 0.34 | 82.5 | 40 | 75 | 66.7 |
| Qwen3-32B | 0.26 | 0.95 | 0.22 | 68.3 | 30 | 55 | 25.0 |
| Qwen2.5-72B-Instruct | 0.26 | 0.60 | 0.24 | 70.0 | 70 | 80 | 56.7 |
| Qwen2.5-32B-Instruct | 0.36 | 0.65 | 0.12 | 53.3 | 50 | 50 | 48.3 |
| Llama-3.1-70B-Instruct | 0.52 | 1.80 | 0.12 | 68.3 | 60 | 80 | 50.0 |
| Llama-Nemo-49B | 0.72 | 1.77 | 0.11 | 65.0 | 30 | 55 | 30.5 |
| Gemma-2-27b-it | 0.19 | 1.65 | 0.43 | 66.7 | 75 | 70 | 35.0 |
| Phi-4-14B | 0.28 | 1.65 | 0.15 | 70.0 | 60 | 80 | 38.3 |
| OLMo2-32B-Instruct | 0.19 | 1.05 | 0.07 | 63.3 | 15 | 30 | 11.7 |
| *Domain Expert Models* | | | | | | | |
| Ether0 | Failed | 0.35 | Failed | Failed | 94 | 76 | 78 |
| BioMedGPT-7B | 1.6 | 2.43 | 0.18 | 53.3 | 10 | 12 | 10 |
| BioMistral-7B | 1.0 | 1.85 | 0.04 | 32.5 | 0 | 10 | 0 |
| *Latent Chemical LLMs* | | | | | | | |
| **LatentChem (Ours)** | **0.09** | **0.15** | **0.81** | **96.7** | **71** | **84** | **61** |

*Table 9.* Comparison with results reported on the ChemCoTBench Molecule Optimization task. The optimized targets are categorized into physicochemical properties (QED, LogP, solubility) and protein activity-related properties (JNK3, DRD2, GSK-3$\beta$), with the latter posing greater challenges to the model's chemical knowledge and reasoning capabilities. $\Delta$ is the mean property improvement, where a negative $\Delta$ indicates that most optimizations are property degradations. SR% is the success rate that brings property increase.

| Models | LogP | | Solubility | | QED | | DRD2 | | JNK3 | | GSK3-$\beta$ | |
|---|---|---|---|---|---|---|---|---|---|---|---|---|
| | $\Delta$ | SR% | $\Delta$ | SR% | $\Delta$ | SR% | $\Delta$ | SR% | $\Delta$ | SR% | $\Delta$ | SR% |
| *W/ Thinking* | | | | | | | | | | | | |
| Gemini-2.5-pro-think | -0.22 | 76 | 1.06 | 70 | 0.28 | 84 | 0.36 | 74 | -0.02 | 35 | 0.06 | 68 |
| Claude3.7-sonnet-think | 0.41 | 80 | 0.37 | 75 | 0.12 | 73 | 0.17 | 63 | 0.01 | 49 | 0.02 | 57 |
| DeepSeek-R1 | 0.47 | 69 | 0.80 | 80 | 0.17 | 72 | 0.12 | 62 | -0.02 | 29 | 0.01 | 41 |
| o3-mini@20250103 | 0.26 | 59 | 0.81 | 85 | 0.21 | 86 | 0.19 | 69 | -0.03 | 23 | 0.01 | 45 |
| o1-mini@20240912 | -0.42 | 52 | 1.78 | 95 | 0.07 | 70 | -0.03 | 37 | -0.10 | 15 | -0.08 | 31 |
| Qwen3-235B-A22B-think | 0.05 | 40 | 0.20 | 40 | 0.02 | 24 | 0.03 | 31 | -0.01 | 23 | 0.01 | 31 |
| Qwen3-32B-think | -0.01 | 1 | 0.13 | 19 | 0.01 | 9 | 0.0 | 4 | -0.02 | 3 | -0.02 | 6 |
| Llama-Nemo-49B-think | -0.24 | 7 | 0.25 | 25.2 | 0.10 | 41 | 0.03 | 29.9 | -0.02 | 6 | -0.01 | 11.2 |
| *W/o Thinking* | | | | | | | | | | | | |
| GPT-4o@20241120 | -0.09 | 37 | 0.92 | 80 | 0.13 | 70 | 0.07 | 48 | -0.02 | 30 | -0.00 | 39 |
| DeepSeek-V3 | 0.09 | 33 | 0.57 | 92 | 0.08 | 46 | 0.03 | 28 | 0.00 | 18 | -0.01 | 29 |
| Gemini-2.0-flash | 0.37 | 72 | 0.28 | 58 | 0.13 | 79 | 0.15 | 63 | -0.02 | 34 | 0.01 | 38 |
| Qwen3-235B-A22B | 0.03 | 21 | 0.18 | 45 | 0.07 | 34 | 0.04 | 26 | -0.01 | 18 | 0.02 | 25 |
| Qwen3-32B | 0.0 | 2 | 0.08 | 20 | 0.02 | 14 | -0.01 | 6 | -0.02 | 6 | -0.02 | 5 |
| Qwen2.5-72B-Instruct | -0.03 | 41 | 0.34 | 59 | 0.07 | 57 | 0.04 | 40 | -0.02 | 26 | -0.00 | 40 |
| Qwen2.5-32B-Instruct | 0.15 | 44 | 0.49 | 65 | 0.09 | 54 | 0.05 | 32 | -0.02 | 19 | 0.01 | 31 |
| Llama-3.1-70B-Instruct | 0.02 | 35 | 0.72 | 81 | 0.15 | 61 | -0.00 | 31 | -0.01 | 30 | 0.01 | 40 |
| Llama-Nemo-Super-49B | -0.01 | 24 | 0.34 | 40 | 0.08 | 43 | -0.00 | 16 | -0.00 | 15 | 0.01 | 27 |
| Gemma-2-27b-it | 0.01 | 31 | 0.39 | 69 | 0.07 | 56 | -0.02 | 15 | -0.00 | 16 | -0.00 | 17 |
| Phi-4-14B | -0.26 | 44 | 0.22 | 53 | 0.17 | 74 | -0.02 | 18 | -0.03 | 14 | -0.00 | 22 |
| OLMo2-32B-Instruct | -1.71 | 11 | 1.21 | 46 | 0.08 | 40 | -0.05 | 7 | -0.03 | 8 | -0.02 | 12 |
| *Domain Expert Models* | | | | | | | | | | | | |
| Ether0 | 0.0 | 0 | 0.0 | 0 | 0.0 | 0 | 0.0 | 0 | 0.0 | 0 | 0.0 | 0 |
| BioMedGPT-7B | -0.36 | 17 | 0.25 | 63 | -0.29 | 7 | -0.09 | 5 | -0.11 | 6 | -0.08 | 1 |
| BioMistral-7B | 0.01 | 1 | 0.24 | 6 | 0.0 | 0 | 0.0 | 1 | -0.01 | 1 | -0.01 | 0 |
| *Latent Chemical LLMs* | | | | | | | | | | | | |
| **LatentChem (Ours)** | **1.37** | **96** | **1.53** | **89** | **0.18** | **83** | **0.26** | **74** | **0.08** | **60** | **0.17** | **82** |

*Table 10.* Comparison with results reported on the ChemCoTBench Chemical Reaction task, restricted to forward prediction (Fwd$_{major}$: major-product prediction, and Fwd$_{by}$: by-product prediction). FTS: molecule fingerprint similarity with reference.

| Models | Fwd $_{major}$ | | Fwd $_{by}$ | |
|---|---|---|---|---|
| | Top-1 | FTS↑ | Top-1 | FTS↑ |
| *W/ Thinking* | | | | |
| Gemini-2.5-pro-think | 0.72 | 0.89 | 0.20 | 0.51 |
| Claude3.7-sonnet-think | 0.73 | 0.87 | 0.25 | 0.31 |
| DeepSeek-R1 | 0.48 | 0.71 | 0.21 | 0.45 |
| o3-mini@20250103 | 0.52 | 0.71 | 0.20 | 0.27 |
| o1-mini@20240912 | 0.26 | 0.31 | 0.11 | 0.17 |
| Qwen3-235B-A22B-think | 0.03 | 0.54 | 0.0 | 0.07 |
| Qwen3-32B-think | 0.11 | 0.33 | 0.09 | 0.18 |
| Llama-Nemo-49B-think | 0.09 | 0.18 | 0.04 | 0.18 |
| *W/o Thinking* | | | | |
| GPT-4o@20241120 | 0.28 | 0.58 | 0.04 | 0.20 |
| DeepSeek-V3 | 0.36 | 0.62 | 0.04 | 0.30 |
| Gemini-2.0-flash | 0.19 | 0.56 | 0.01 | 0.07 |
| Qwen3-235B-A22B | 0.04 | 0.57 | 0.0 | 0.06 |
| Qwen3-32B | 0.06 | 0.57 | 0.0 | 0.13 |
| Qwen2.5-72B-Instruct | 0.04 | 0.49 | 0.0 | 0.13 |
| Qwen2.5-32B-Instruct | 0.01 | 0.43 | 0.0 | 0.12 |
| Llama-3.1-70B-Instruct | 0.02 | 0.35 | 0.0 | 0.08 |
| Llama-Nemo-49B | 0.04 | 0.40 | 0.0 | 0.08 |
| Gemma-2-27b-it | 0.01 | 0.55 | 0.0 | 0.04 |
| Phi-4-14B | 0.01 | 0.27 | 0.03 | 0.10 |
| OLMo2-32B-Instruct | 0.0 | 0.10 | 0.0 | 0.07 |
| Text+Chem T5 | 0.44 | 0.74 | 0.0 | 0.07 |
| *Latent Chemical LLMs* | | | | |
| **LatentChem (Ours)** | **0.17** | **0.52** | **0.08** | **0.17** |

## K. Inference Latency and Budget Analysis

We additionally measure model-side wall-clock latency using synchronized CUDA events under `bfloat16` inference with PyTorch 2.9.0+cu128 and CUDA 12.8. Evaluation is run on 8 NVIDIA H100 GPUs with process-level sample sharding, where each process hosts a full model replica on a single GPU. Latency is evaluated at batch size 1, and the first 3 runs after each model load are discarded as warm-up. All other inference settings are kept identical to those used in the main experiments. The timed region covers the full model-side inference pass from prompt construction to the end of autoregressive decoding, and all reported latency values are computed as the mean over the per-sample measurements collected after warm-up.

*Table 11.* Model-side wall-clock relative speed by task group.

| Task group | Relative speed |
|---|---|
| Reaction | 10.1× |
| Molecule optimization | 6.4× |
| Molecule editing | 4.2× |
| Molecule understanding | 3.9× |
| Molecule description | 0.9× |

We also compare the model with zero latent budget ($T = 0$) against the explicit CoT baseline on the same optimization and scaffold-oriented understanding metrics. The $T = 0$ model is largely comparable to CoT on optimization, indicating fallback to textual reasoning, though some topology-sensitive understanding tasks still benefit from latent budget.

*Table 12.* Comparison between zero latent budget ($T = 0$) and the explicit CoT baseline. Success rate (SR%) is reported for optimization tasks, scaffold similarity for Murcko, and correct rate for Ring-sys.

| Method | Molecule optimization | | | | | | Molecule understanding | |
|---|---|---|---|---|---|---|---|---|
| | LogP SR% | Solubility SR% | QED SR% | DRD2 SR% | JNK3 SR% | GSK3-$\beta$ SR% | Murcko | Ring-sys |
| $T = 0$ | 78 | 85 | 83 | 74 | 44 | 70 | 0.64 | 65 |
| CoT baseline | 77 | 86 | 82 | 70 | 44 | 67 | 0.65 | 83 |

## L. GRPO Details and Reward Diagnostics

GRPO samples ChemCoTDataset tasks uniformly, with correctness checked by task-specific benchmark oracles. We do not use task-specific reward calibration. Table 13 reports per-task reward hit rates, and Table 14 reports format-adherence rates across evaluation benchmarks. Except for the untuned text-only baseline, format adherence remains above 97% on most benchmarks. A longer 3-epoch GRPO run slightly decreases open-ended performance, slightly improves closed-ended performance, and improves token efficiency. Because RL-based post-training can be non-monotonic with training duration, additional optimization does not necessarily improve task performance; we therefore use the 1-epoch GRPO model as a conservative practical stopping point in this work, while leaving a fuller characterization of GRPO instability and over-optimization to future work.

*Table 13.* Reward-hit diagnostics for LatentChem.

| Task | Subtask | Reward hit rate | | |
|---|---|---|---|---|
| | | Format | Validity | Correctness |
| Understanding | FG Count | 1.00 | 1.00 | 0.92 |
| Understanding | Ring Count | 1.00 | 1.00 | 0.90 |
| Understanding | Murcko Scaffold | 1.00 | 0.93 | 0.68 |
| Understanding | Ring System | 1.00 | 1.00 | 0.81 |
| Reaction | Reaction | 0.92 | 0.87 | 0.65 |
| Editing | Add | 1.00 | 0.91 | 0.64 |
| Editing | Delete | 1.00 | 0.88 | 0.71 |
| Editing | Substitute | 1.00 | 0.89 | 0.82 |
| Optimization | LogP | 1.00 | 0.89 | 0.82 |
| Optimization | Solubility | 1.00 | 0.89 | 0.82 |
| Optimization | QED | 1.00 | 0.93 | 0.75 |
| Optimization | DRD2 | 1.00 | 0.91 | 0.70 |
| Optimization | JNK3 | 1.00 | 0.87 | 0.53 |
| Optimization | GSK3-$\beta$ | 1.00 | 0.88 | 0.73 |

*Table 14.* Format-adherence rates on the evaluation benchmarks.

| Method | Benchmark | | | |
|---|---|---|---|---|
| | ChemCoTBench | Mol-Instructions | ChemLLMBench | ChEBI-20 |
| Qwen-3-8B | 53.75% | 65.08% | 61.17% | 62.18% |
| Qwen-3-8B (SFT) | 98.88% | 99.80% | 99.83% | 98.97% |
| Stage 1 | 100.00% | 99.95% | 97.83% | 99.70% |
| Stage 1+2 | 98.03% | 97.95% | 99.00% | 99.64% |
| Stage 1+2+4 | 99.74% | 99.78% | 99.00% | 99.94% |
| Coconut-Chem | 98.62% | 97.45% | 97.83% | 95.78% |
| LatentChem | 99.87% | 99.35% | 99.83% | 98.00% |

## M. Additional Baselines and Training Variants

Table 15 reports additional evaluations of Qwen-3-8B (SFT+GRPO), Stage 1+4, Stage 1+2+3, joint Stage-4 training, and 3-epoch GRPO variants using the same non-tie win-rate metric against the CoT baseline. For the joint Stage-4 variant, the latent modules are unfrozen and optimized jointly with the LLM; this variant is weaker than the frozen-latent LatentChem design in overall performance and in average reasoning-step overhead reduction. We interpret this result primarily as an optimization issue rather than evidence that the learned latent representations are brittle: the latent modules are lightweight relative to the backbone, making joint GRPO optimization harder. This supports our staged design, which first learns the latent interface with a frozen backbone and then keeps the latent modules fixed during RL.

*Table 15.* Additional baselines and Stage-4 variants using non-tie win rate against the explicit CoT baseline.

| Method | ChemCoTBench | | | Mol-Instructions | | | ChemLLMBench | | | ChEBI-20 |
| --- | --- | --- | --- | --- | --- | --- | --- | --- | --- | --- |
| | Open | Closed | All | Open | Closed | All | Open | Closed | All | Open |
| Qwen-3-8B (SFT+GRPO) | 36.53 | 38.46 | 36.97 | 76.32 | 45.59 | 47.71 | 82.65 | 46.55 | 49.43 | 80.49 |
| Stage 1+4 | 49.23 | 43.04 | 47.83 | 67.00 | 49.28 | 50.38 | 64.84 | 50.23 | 51.34 | 65.53 |
| Stage 1+2+3 | 37.15 | 20.25 | 33.33 | 52.83 | 42.06 | 42.73 | 58.62 | 44.59 | 45.62 | 59.82 |
| Joint Stage 4 | 55.65 | 45.75 | 53.41 | 67.29 | 48.15 | 49.35 | 68.54 | 52.13 | 53.37 | 73.53 |
| LatentChem (GRPO 3 epochs) | 49.21 | 49.73 | 49.31 | 69.49 | 52.91 | 54.01 | 84.04 | 57.18 | 59.35 | 81.58 |

*Table 16.* Reasoning-step overhead reduction of joint Stage-4 training compared with the frozen-latent LatentChem design. The mean is computed across the listed task groups.

| Task group | Reasoning-step overhead reduction | |
| --- | --- | --- |
| | Joint Stage 4 | LatentChem |
| Reaction | $4.0\times$ | $29.9\times$ |
| Molecule optimization | $0.8\times$ | $28.4\times$ |
| Molecule editing | $18.4\times$ | $13.4\times$ |
| Molecule understanding | $7.2\times$ | $10.2\times$ |
| Molecule description | $2.4\times$ | $5.4\times$ |
| Mean | $6.55\times$ | $10.84\times$ |

# N. ChemUpdater Ablation on Topology-Sensitive Tasks

To directly test whether ChemUpdater contributes structure-aware refinement, we evaluate topology-sensitive scaffold tasks. As shown in Table 17, removing ChemUpdater reduces Murcko scaffold similarity from 0.81 to 0.67 and ring-system accuracy from 97 to 80.

*Table 17.* Topology-sensitive ChemUpdater ablation.

| Model | Topology-sensitive task | |
| --- | --- | --- |
| | Murcko ↑ | Ring-sys ↑ |
| w/o latent thinking | 0.65 | 83 |
| w/o latent projector | 0.69 | 88 |
| w/o ChemUpdater | 0.67 | 80 |
| LatentChem | 0.81 | 97 |

## O. Counterfactual Alignment Ablation

A direct Stage-1 ablation suggests that $\mathcal{L}_{CF}$ is not essential for the main gains; we therefore view it as an optional grounding regularizer. Table 18 reports the full Open/Closed/All breakdown for the Stage-1 model without $\mathcal{L}_{CF}$.

*Table 18.* Stage-1 counterfactual-alignment ablation using non-tie win rate against Stage 1.

| Model | ChemCoTBench | | | Mol-Instructions | | | ChemLLMBench | | | ChEBI-20 |
|---|---|---|---|---|---|---|---|---|---|---|
| | Open | Closed | All | Open | Closed | All | Open | Closed | All | Open |
| Stage 1 w/o $\mathcal{L}_{CF}$ vs. Stage 1 | 51.62 | 43.87 | 49.38 | 62.77 | 50.25 | 51.09 | 62.35 | 49.31 | 50.32 | 72.28 |

## P. Dataset Overlap and Generalization Protocol

ChemCoTDataset and ChemCoTBench are separate resources rather than train/test splits of the same 14k set. Based on exact field matches of task type, input molecule, and final answer, 8 overlapping ChemCoTBench records (0.71%) were detected; no overlaps were detected on the other benchmarks. GRPO rollouts use the same ChemCoTDataset prompt pool as SFT, so the generalization evidence is transfer to disjoint evaluation benchmarks rather than unseen RL prompts.

## Q. Win/Lose/Tie Breakdown and Robustness

Tables 20, 21, 22, and 23 are win/lose/tie breakdown. Table 19 reports repeated evaluation across seeds 2026–2075, where the stable mean±std non-tie win rates are consistent with Table 3.

*Table 19.* Non-tie win rates across seeds 2026–2075.

| Seeds | ChemCoTBench | | | Mol-Instructions | | | ChemLLMBench | | | ChEBI-20 |
| | Open | Closed | All | Open | Closed | All | Open | Closed | All | Open |
| --- | --- | --- | --- | --- | --- | --- | --- | --- | --- | --- |
| 2026–2075 | 64.94±1.31 | 44.52±2.40 | 60.74±1.21 | 71.14±1.23 | 48.12±0.43 | 49.62±0.40 | 89.54±3.26 | 52.23±1.51 | 55.10±1.41 | 85.69±0.49 |

*Table 20.* Win/lose/tie breakdown on ChemCoTBench.

| Method | Open | | | Closed | | | All | | |
| | Win | Lose | Tie | Win | Lose | Tie | Win | Lose | Tie |
| --- | --- | --- | --- | --- | --- | --- | --- | --- | --- |
| Qwen-3-8B | 11.05% | 68.03% | 20.92% | 1.52% | 52.65% | 45.83% | 5.58% | 59.20% | 35.22% |
| Qwen-3-8B (SFT) | 19.75% | 50.58% | 29.67% | 7.06% | 27.17% | 65.77% | 13.63% | 39.28% | 47.09% |
| Stage 1 | 29.33% | 37.58% | 33.08% | 11.17% | 19.21% | 69.62% | 20.57% | 28.72% | 50.71% |
| Stage 1+2 | 25.92% | 41.67% | 32.42% | 8.31% | 20.91% | 70.78% | 17.42% | 31.65% | 50.93% |
| Coconut-Chem | 24.58% | 46.25% | 29.17% | 9.07% | 25.41% | 65.52% | 17.16% | 36.27% | 46.57% |
| LatentChem | 39.92% | 22.67% | 37.42% | 13.06% | 13.06% | 73.88% | 26.96% | 18.03% | 55.00% |

*Table 21.* Win/lose/tie breakdown on Mol-Instructions.

| Method | Open | | | Closed | | | All | | |
| | Win | Lose | Tie | Win | Lose | Tie | Win | Lose | Tie |
| --- | --- | --- | --- | --- | --- | --- | --- | --- | --- |
| Qwen-3-8B | 54.07% | 38.12% | 7.81% | 15.18% | 54.22% | 30.60% | 16.87% | 53.52% | 29.61% |
| Qwen-3-8B (SFT) | 71.44% | 26.25% | 2.30% | 23.55% | 42.01% | 34.45% | 25.73% | 41.29% | 32.98% |
| Stage 1 | 55.61% | 32.67% | 11.72% | 32.43% | 31.80% | 35.77% | 33.49% | 31.84% | 34.68% |
| Stage 1+2 | 42.24% | 41.64% | 16.12% | 27.01% | 34.82% | 38.17% | 27.72% | 35.14% | 37.14% |
| Coconut-Chem | 53.28% | 38.69% | 8.03% | 26.08% | 40.04% | 33.87% | 27.30% | 39.98% | 32.72% |
| LatentChem | 66.63% | 25.92% | 7.45% | 30.39% | 32.44% | 37.17% | 32.01% | 32.15% | 35.84% |

*Table 22.* Win/lose/tie breakdown on ChemLLMBench.

| Method | Open | | | Closed | | | All | | |
|---|---|---|---|---|---|---|---|---|---|
| | Win | Lose | Tie | Win | Lose | Tie | Win | Lose | Tie |
| Qwen-3-8B | 61.02% | 30.51% | 8.47% | 13.28% | 54.50% | 32.21% | 15.72% | 53.28% | 31.00% |
| Qwen-3-8B (SFT) | 79.00% | 16.00% | 5.00% | 24.75% | 41.23% | 34.03% | 27.79% | 39.81% | 32.40% |
| Stage 1 | 63.00% | 29.00% | 8.00% | 34.13% | 33.38% | 32.50% | 35.83% | 33.12% | 31.05% |
| Stage 1+2 | 52.00% | 35.00% | 13.00% | 29.04% | 35.33% | 35.63% | 30.34% | 35.31% | 34.35% |
| Coconut-Chem | 69.47% | 23.16% | 7.37% | 28.72% | 38.91% | 32.37% | 30.91% | 38.06% | 31.03% |
| LatentChem | 83.84% | 12.12% | 4.04% | 34.89% | 31.10% | 34.00% | 37.60% | 30.05% | 32.34% |

*Table 23.* Win/lose/tie breakdown on ChEBI-20.

| Method | Open | | |
|---|---|---|---|
| | Win | Lose | Tie |
| Qwen-3-8B | 71.24% | 23.93% | 4.83% |
| Qwen-3-8B (SFT) | 78.32% | 18.55% | 3.13% |
| Stage 1 | 59.91% | 32.54% | 7.55% |
| Stage 1+2 | 47.55% | 39.54% | 12.92% |
| Coconut-Chem | 61.09% | 32.07% | 6.84% |
| LatentChem | 80.09% | 13.84% | 6.07% |

