# OpenReview forum: "LatentChem: From Textual CoT to Latent Thinking in Chemical Reasoning"
_ICML.cc/2026/Conference — ICML 2026 regular_

### Official Review · Reviewer_Ehya · 2026-03-06

**Soundness:** 3
**Presentation:** 3
**Significance:** 3
**Originality:** 2
**Overall Recommendation:** 4
**Confidence:** 5

**Summary:**

This paper proposes LatentChem, a chemical reasoning framework that shifts reasoning from explicit textual CoT to continuous latent thought vectors, coupled with a dynamic perception–reasoning loop in which the ChemUpdater repeatedly refines molecular representations during inference. The central empirical claim is that, after Stage-4 reward-only optimization, the model exhibits “spontaneous internalization”: it stops emitting verbose textual CoT and instead performs most intermediate reasoning in latent space. Experimentally, the paper presents broad evaluation across four chemistry benchmarks and shows its strongest gains on open-ended generative tasks, while improvements on closed-ended deterministic tasks are notably narrower and closer to parity with the explicit CoT baseline. Overall, the paper is novel and ambitious, with a distinctive method and strong open-ended results, though some of its broader claims about efficiency and general superiority appear stronger than the current evidence fully supports.

**Compliance With Llm Reviewing Policy:**

Affirmed.

**Final Justification:**

I maintain my overall recommendation of Weak Accept. This is a technically solid paper with a meaningful chemistry-specific extension of latent reasoning, particularly the dynamic perception-reasoning loop through ChemUpdater. The empirical evidence is most convincing on open-ended generative tasks, and the mechanistic analyses are useful. At the same time, the closed-ended results are more mixed, and some of the broader claims in the original draft were stronger than the evidence fully supported.

The rebuttal addressed my main concerns partially but constructively. The added wall-clock measurements and follow-up clarification substantially improve the efficiency claim, assuming the final paper clearly separates reasoning-step/token overhead from actual runtime efficiency. The rebuttal also strengthens the interpretation of spontaneous internalization and the support for ChemUpdater. These responses reinforce rather than fundamentally change my prior assessment: I view this as a good paper with clear technical value, but one whose claims should be calibrated carefully in the final version.

**Key Questions For Authors:**

Please see the weaknesses above; my main questions concern (i) the fairness of the efficiency claim, (ii) the mixed results on closed-ended tasks, (iii) the interpretation of spontaneous internalization, and (iv) the lack of diagnostic ablations for ChemUpdater.

**Limitations:**

yes

**Strengths And Weaknesses:**

**Strengths:**

* **The method offers a substantive technical contribution rather than a mere rebranding of latent tokens.**
  LatentChem goes beyond introducing latent tokens by incorporating a **dynamic perception–reasoning loop**. Through the ChemUpdater, the model can repeatedly re-query and refine molecular representations during reasoning, which is more principled than treating molecular features as static context.

* **The empirical results are particularly strong on open-ended chemical generation and optimization tasks.**
  The paper presents a fairly comprehensive evaluation across four benchmarks, and the gains are most convincing on open-ended generative settings. In particular, the method shows consistently strong performance across a broad set of open-ended metrics, which makes the empirical case compelling.

* **The paper includes meaningful mechanistic analysis rather than only reporting benchmark gains.**
  Beyond headline results, the authors provide analyses of latent dynamics, task-specific clustering, and phenomena such as latent budget and causal noise, all of which help support the claim that the latent states are carrying non-trivial reasoning signals rather than acting as placeholders.

**Weaknesses:**


* **The efficiency claim is overstated relative to the reported metric.**
  The paper measures “efficiency” via a step-count proxy that compares explicit CoT length against LatentChem’s latent-plus-textual steps, but this does not directly establish improvements in wall-clock latency, FLOPs, or memory. As currently presented, the evidence supports reduced reasoning overhead more than true system-level efficiency.

* **Results on closed-ended tasks are mixed and under-discussed.**
  The paper itself notes that gains are strongest on open-ended generative tasks, while margins on deterministic closed-ended tasks are much narrower. This weakens the broader claim that latent reasoning is uniformly advantageous for chemical reasoning.

* **The evidence for “spontaneous internalization” does not fully rule out a shorter-output bias.**
  In Stage 4, explicit CoT supervision is removed and optimization is driven only by final-answer rewards, so part of the observed internalization may reflect a preference for shorter rewarded trajectories rather than genuinely better latent reasoning. The current analyses are suggestive, but not yet decisive on this causal point.

* **The ChemUpdater ablation is not diagnostic enough for the claimed mechanism.**
  The paper claims that ChemUpdater enables dynamic structure-aware refinement during reasoning, but the ablation evidence is limited to broad generative outcomes. To support the mechanism claim, the study should include tasks requiring precise topological or substructure tracking.

---

> ### Author Rebuttal · Authors · 2026-03-31
>
> **W1 / Key Q(i): the fairness of the efficiency claim**
>
> We agree that some of the current wording in the paper conflates token overhead with speed / inference efficiency, which is imprecise. Beyond the step ratio, we now additionally measured end-to-end wall-clock time and observed a 5.96× overall speedup. Specifically, LatentChem is faster on reaction, optimization, editing, and understanding tasks, while description is a little slower because runtime there is dominated by long textual outputs, i.e., the total runtime in this setting is affected more by the length of the final answer. Overall, from a wall-clock perspective, the gain in reasoning efficiency is still substantial. In the revision, we will correct all places where token overhead could be misread as true system-level efficiency; specifically, our revision would be:
> - “average speedup” -> “average reduction in reasoning step overhead” (line 32)
> - “speed” -> “token efficiency” (line 60).
>
> |Task group|Relative speed|
> |-|-:|
> |Reaction|10.1×|
> |Mol.optimization|6.4×|
> |Mol.editing|4.2×|
> |Mol.understanding|3.9×|
> |Mol.description|0.9×|
>
> **W2 / Key Q(ii): mixed results on closed-ended tasks**
>
> We agree the benefit is task-dependent at the subtask level. Our claim is not uniform superiority on every chemical task: on open-ended tasks, LatentChem improves both effectiveness and efficiency; on closed-ended tasks, it remains broadly comparable in effectiveness while substantially reducing reasoning step overhead. Taken together, these results support our broader claim that latent reasoning is overall advantageous as a reasoning paradigm for chemical tasks, even though the gains are not uniform across all individual subtasks. We will make this logic more explicit in the revision.
>
> **W3 / Key Q(iii): the interpretation of spontaneous internalization**
>
> We appreciate this concern, and we agree that the wording should be more precise; however, our current results are not fully explained by a simple brevity-bias explanation. In Stage 4, the reward contains no term for brevity or CoT omission; it only evaluates format, validity, and task correctness (Sec. 3.3). If outcome-only RL simply favored shorter outputs, then the explicit-CoT Stage 1+2+4 baseline trained under the same objective should also collapse to near-zero CoT, which it does not (Fig. 8). Moreover, in LatentChem, explicit CoT re-emerges when the latent budget is restricted (Fig. 6), indicating a capacity-dependent tradeoff between latent and textual reasoning rather than a generic short-output bias. More generally, we acknowledge that “spontaneous internalization” is our descriptive term rather than a standardized technical term, so it may admit different interpretations. In the revision, we will define it explicitly at first use as “self-selected under outcome-only optimization,” rather than implying emergence without architectural design or objective-independent internalization.
>
> **W4 / Key Q(iv): the lack of diagnostic ablations for ChemUpdater**
>
> Thank you for this helpful suggestion. To directly test the claimed mechanism, we added topology-sensitive scaffold evaluations. Removing ChemUpdater reduces Murcko scaffold similarity from 0.81 to 0.67 and ring-system accuracy from 97 to 80. These results more directly support the claim that ChemUpdater contributes structure-aware refinement.
>
> | Model                | Murcko ↑ | Ring-sys ↑ |
> | -------------------- | -------: | ---------: |
> | w/o latent thinking  |     0.65 |         83 |
> | w/o latent projector |     0.69 |         88 |
> | w/o ChemUpdater      |     0.67 |         80 |
> | LatentChem           |     0.81 |         97 |

---

> > ### Author Rebuttal · Reviewer_Ehya · 2026-04-03
> >
> > Thank you for the detailed rebuttal. My concerns are partially resolved.
> >
> > I still have a follow-up question. On efficiency, could the revision report the full wall-clock measurement protocol and clearly separate reduced reasoning-step/token overhead from actual runtime efficiency throughout the paper?

---

> > > ### Author Response · Authors · 2026-04-04
> > >
> > > Thank you for the helpful follow-up. In the revision, we will report the full wall-clock measurement protocol and clearly separate reasoning-step/token-overhead reduction from actual runtime efficiency. Specifically, our protocol is as follows:
> > >
> > > > We report model-side wall-clock latency measured using synchronized CUDA events under bfloat16 inference (PyTorch 2.9.0+cu128, CUDA 12.8). Evaluation is run on 8 NVIDIA H100 GPUs with process-level sample sharding, where each process hosts a full model replica on a single GPU. Latency is evaluated at batch size 1, and the first 3 runs after each model load are discarded as warm-up. All other inference settings are kept identical to those used in the main experiments. The timed region covers the full model-side inference pass from prompt construction to the end of autoregressive decoding. All reported latency values are computed as the mean over the per-sample measurements collected after warm-up.
> > >
> > > We will report the wall-clock result in Sec. 5.2 ("Inference efficiency"), together with a brief description of the measurement setup, and provide the full protocol in the appendix of the revised paper.
> > >
> > > To avoid any remaining ambiguity between reductions in reasoning-step overhead and actual runtime efficiency, we will revise all relevant "[number]×" statements in the current manuscript so that they explicitly refer to reasoning-step/token overhead where appropriate. Where needed, we will also report the measured 5.96× wall-clock speedup in the same sentence. Concretely, the revision will include the following wording changes:
> > >
> > > - **Line 032**: "10.84× average speedup" **->** "10.84× average reduction in reasoning step overhead (5.96× wall-clock speedup)"
> > > - **Lines 103-106**: "LatentChem achieves a dramatic efficiency gain, reducing the reasoning token overhead by an average of 10.84× across all benchmarks. In molecule optimization and reaction tasks, this reduction factor even exceeds 28×." **->** "LatentChem achieves a dramatic reduction in reasoning step overhead, averaging 10.84× across all benchmarks (with a measured 5.96× wall-clock speedup overall). In molecule optimization and reaction tasks, this reduction factor even exceeds 28×."
> > > - **Lines 376-378**: "As shown in Figure 8, by compressing verbose linguistic thought, LatentChem reduces this overhead by factors ranging from 5.4× to 29.9× (average 10.84×)." **->** "As shown in Figure 8, by compressing verbose linguistic thought, LatentChem reduces reasoning step overhead by factors ranging from 5.4× to 29.9× (average 10.84×, with a measured 5.96× wall-clock speedup)."
> > > - **Line 421**: "LatentChem achieves up to 29.9× efficiency on reaction tasks" **->** "LatentChem achieves up to 29.9× reduction in reasoning step overhead on reaction tasks"
> > > - **Line 422**: "a 10.84× efficiency gain" **->** "a 10.84× reduction in reasoning step overhead (5.96× wall-clock speedup)"
> > >
> > > We believe these revisions will remove the remaining ambiguity and clearly separate reasoning-step/token-overhead reduction from actual runtime efficiency throughout the paper.

---

### Official Review · Reviewer_5Cfh · 2026-03-08

**Soundness:** 3
**Presentation:** 4
**Significance:** 3
**Originality:** 3
**Overall Recommendation:** 4
**Confidence:** 3

**Summary:**

This paper investigates whether explicit textual Chain-of-Thought (CoT) reasoning is necessary for complex chemical reasoning in large language models (LLMs). The authors argue that forcing chemical logic—intrinsically continuous and structural—into discrete natural language tokens creates a “modality mismatch” that may limit reasoning efficiency and performance.

To explore this hypothesis, they propose LatentChem, a framework that decouples chemical reasoning from linguistic generation. Instead of generating intermediate reasoning steps as text, LatentChem introduces a latent thinking phase composed of continuous thought vectors. These latent states interact dynamically with molecular representations through a ChemUpdater module, which allows the model to iteratively refine its structural understanding during reasoning. A latent projector enables recurrent reasoning in embedding space without passing through discrete token generation.

The model is trained through a multi-stage protocol: supervised alignment with molecular inputs, supervised CoT training, activation of latent reasoning modules, and finally reinforcement learning (via GRPO) with rewards focused only on answer correctness and format. During this final stage, the model exhibits an emergent behavior termed spontaneous internalization, in which it abandons verbose textual CoT and performs reasoning almost entirely in latent space while still improving task performance.

Empirically, LatentChem is evaluated on four chemical benchmarks spanning molecule understanding, editing, optimization, description, and reaction prediction. The results show that latent reasoning achieves superior or competitive accuracy compared to strong explicit CoT baselines, particularly on open-ended generative tasks such as molecule optimization. Additionally, it significantly reduces inference cost, achieving large reductions in reasoning token usage (averaging over 10× speedup).

The paper’s main contributions are:

1.Formulating the “modality mismatch” hypothesis for chemical reasoning in LLMs.
2.Introducing the LatentChem architecture that enables continuous latent reasoning with dynamic molecular refinement.
3.Demonstrating the emergent internalization of explicit CoT into latent computation under outcome-based optimization.
4.Empirically showing that latent reasoning improves both performance and efficiency across diverse chemical reasoning benchmarks.
Overall, the work presents latent reasoning as a viable alternative to explicit textual CoT for structure-heavy scientific tasks, suggesting that continuous internal computation may better align with the nature of chemical reasoning.

**Compliance With Llm Reviewing Policy:**

Affirmed.

**Key Questions For Authors:**

1.How do you distinguish between "emergent" behavior and "directly incentivized but not explicitly supervised" behavior? The GRPO loss doesn't penalize omitting CoT, which is functionally equivalent to encouraging brevity.Have you conducted experiments where CoT generation is also rewarded (e.g., adding a KL penalty to maintain linguistic reasoning)? Does internalization still occur?

2.You state that ChemCoTBench's test set  is "held-out" from training, but both the training data  and test set appear to be derived from the same collection methodology and potentially overlapping source datasets. Additionally, Table 2 shows that LatentChem significantly outperforms baselines on ChemCoTBench but shows much weaker or tied performance on external benchmarks.

3.The claimed 10.8× speedup is measured in token count rather than actual inference latency. Since ChemUpdater introduces additional forward passes and cross-attention operations, reporting wall-clock time or FLOPs would strengthen the efficiency claim.

**Limitations:**

The paper discusses some technical limitations (e.g., domain specificity to chemistry, reduced interpretability due to latent reasoning, and lack of validation beyond the evaluated benchmarks), but the discussion could be strengthened in the following ways:
1.Interpretability and Scientific Reliability
While the paper acknowledges that explicit CoT disappears, it does not fully explore the implications for scientific auditability and trust.
2. Dual-Use Risks in Chemical Generation
Given that the system improves efficiency and performance in molecule optimization and reaction prediction, it could potentially lower barriers to designing harmful compounds.

**Strengths And Weaknesses:**

1. Soundness

Strengths:

(1) Clear hypothesis and architectural alignment.
The paper formulates a concrete and testable hypothesis—the modality mismatch between continuous chemical structure space and discrete language tokens. The proposed LatentChem architecture (latent thinking vectors + ChemUpdater + latent projector) is directly motivated by this hypothesis. The method design is coherent with the stated problem. The formal efficiency analysis (Appendix A) is impressive, providing a geometric framework via the curvature-resolution theorem (Theorem A.9)

(2)  Training protocol is well-motivated.The training protocol is methodologically sound with well-justified design choices (counterfactual alignment, progressive freezing strategies).The staged training (SFT → CoT exposure → latent module activation → RL with outcome-only rewards) is conceptually sound.

(3) Empirical evaluation spans diverse tasks. The experiments cover multiple chemical reasoning settings.The evaluation was comprehensive, the baseline model was appropriately chosen, and the statistical tests were quite rigorous.

Weaknesses:

(1) Limited theoretical grounding. The geometric argument about curvature differences between discrete and continuous spaces is suggestive but not rigorously developed into a formal theoretical result. It reads more as an intuition than a formal proof of efficiency gains.

(2) Interpretability trade-offs are not deeply analyzed. The paper acknowledges sacrificing transparency but doesn't quantify the loss, more discussion of interpretability loss and its practical implications (especially in scientific domains) would strengthen the evaluation.

(3) Generalization Beyond Chemistry. All experiments are chemistry-focused; claims about "scientific reasoning" broadly are not validated.Physics, mathematics, or other domains may not exhibit the same curvature properties.

(4) Scale Limitations. Only evaluated on 8B parameter models (Qwen-3-8B backbone) Unclear whether benefits hold for smaller models or larger models

2. Presentation

Strengths:

(1) Clear narrative arc. The conceptual illustration (Figure 1) immediately conveys the core hypothesis. Progressive revelation,Section 4 (emergent properties) → Section 5 (benchmarking) is pedagogically effective. The three-tier comparison figure (Figure 2) clearly distinguishes LatentChem from CoT and generic latent approaches.

(2) Reproducibility. Exceptional detail in Appendix D (hyperparameters, optimizer settings, learning rates per stage).Dataset preprocessing, metric definitions (Appendix E), and evaluation protocols fully specified.

Weaknesses:

(1) Some concepts would benefit from tighter formalization. Some concepts would benefit from tighter formalization.
Terms such as “latent thinking” and “continuous reasoning” could be more precisely defined (e.g., what constitutes a reasoning step in latent space?). Without formalization, some claims risk sounding metaphorical.

(2) Result Organization, Tables 5-6 (Appendix F) for closed-ended tasks feel disconnected from the main narrative Consider integrating key results into the main text with full details in appendix.

3. Significance

Strengths:

(1) Conceptual Contribution, Challenges the assumption that "thinking = language generation" in LLMs. Provides empirical evidence for the "modality mismatch" hypothesis in scientific domains. The spontaneous internalization phenomenon is genuinely surprising and thought-provoking.

(2) Efficiency gains are practically meaningful. A 10×–30× reduction in reasoning tokens is not merely cosmetic—it directly affects inference cost and latency in real-world systems.

Weaknesses:

(1) Domain specialization. Impact is currently strongest within chemical and structural domains. Broader implications for general reasoning remain speculative.

4. Originality

Strengths:

(1)Novel framing,modality mismatch hypothesis.The idea that CoT is a representational bottleneck for structure-heavy domains is conceptually original and thought-provoking.

(2)Empirical demonstration of spontaneous internalization. The observation that models abandon explicit CoT under outcome-only rewards is a novel empirical contribution.

Weaknesses:

(1)Components are individually known. Latent computation, embedding-space updates, and RL-based reasoning optimization have precedents. The novelty lies more in synthesis and domain application than in entirely new algorithms.

(2) Theoretical Framework Limitations. The curvature-resolution theorem is elegant but qualitative, no quantitative predictions.

---

> ### Author Rebuttal · Authors · 2026-03-31
>
> **Soundness (1) / Originality (2): theoretical grounding is qualitative**
>
> We appreciate this point. Throughout Appendix A, our goal is to provide interpretive analysis under simplifying assumptions, not a formal proof of speedups or a quantitative predictive theory of efficiency. We will further sharpen the wording in the revision to make this framing even clearer.
>
> **Soundness (2) / Limitation (1): interpretability, auditability, trust need deeper analysis**
>
> We will strengthen the limitations discussion on interpretability, especially regarding scientific auditability and trust, by making this trade-off and its practical implications more explicit and by discussing possible directions for selectively externalizing latent reasoning when explanation or verification is needed.
>
> **Soundness (3) / Significance (1): domain specialization**
>
> We appreciate this point and would like to clarify that the claims and empirical scope of this paper are chemistry-specific, and any broader implications beyond chemical reasoning, as discussed in the future-work section, should be viewed as forward-looking. We will revise the discussion to make this scope more explicit.
>
> **Soundness (4)Scale limitations**
>
> We agree and therefore trained matched 4B and 14B versions and evaluated them against their own scale-matched explicit-CoT baselines, using the same non-tie win rate metric (%) as in the main results (Table 2). As shown below, LatentChem remains advantageous overall at both scales across benchmarks.
> |Model||ChemCoTBench|||Mol-Instructions|||ChemLLMBench||ChEBI-20|
> |-|-|-|-|-|-|-|-|-|-|-|
> ||Open|Closed|All|Open|Closed|All|Open|Closed|All|Open|
> |LatentChem-4B|55.41|41.28|52.42|48.37|52.18|51.93|64.29|56.38|57.03|57.07|
> |LatentChem-14B|52.29|49.72|51.80|55.65|51.49|51.81|51.02|55.02|54.69|55.43|
>
> **Presentation (1): some concepts would benefit from tighter formalization**
>
> We agree and will define these terms operationally in the method. Specifically, latent thinking is recurrent generation of continuous hidden states fed back without textual decoding; one latent step is one such update; and continuous reasoning is intermediate computation in latent space rather than via discrete text tokens.
>
> **Presentation (2): result organization**
>
> We will move the key closed-ended summary from Appendix F into the main results/discussion.
>
> **Originality (1): components are individually known**
>
> We would like to clarify our intended contribution framing. The novelty of this work is not merely in combining existing ingredients or domain application, but in establishing and characterizing a chemistry-specific latent reasoning paradigm. In particular, the paper identifies a setting in which latent reasoning is especially effective, and provides empirical evidence and analysis for this advantage relative to explicit CoT. We will revise the framing to make this clearer.
>
> **Key Q1: “emergent” vs directly incentivized brevity**
>
> We would like to clarify that, in our usage, “emergent” refers to a behavior that is not directly incentivized, but is instead self-selected under outcome-only optimization. If outcome-only RL simply favored shorter outputs, then the explicit-CoT Stage 1+2+4 baseline trained under the same objective should also collapse to near-zero CoT, which it does not (Fig. 8). This is not consistent with a simple brevity-bias explanation. Our claim here is narrower: under the current outcome-only setting, the observed shift is better described as self-selected internalization than as directly incentivized brevity. We did not conduct a CoT-preserving control during the rebuttal period, although we agree it would be a useful follow-up. To avoid ambiguity, in the revision we will define “spontaneous internalization” explicitly at first use as “self-selected under outcome-only optimization.”
>
> **Key Q2: train-test curation / performance on external benchmarks**
>
> We would like to clarify two points. First, ChemCoTDataset and ChemCoTBench are separate resources rather than train/test splits of the same 14k set. Using task type + input molecule + final answer, we found only 8 overlapping ChemCoTBench records (0.71%), with no overlaps on the other benchmarks. We agree that broader source-level overlap via shared public databases remains a limitation. We will make this explicit in revision. Second, the external benchmarks show the same task-type-dependent pattern as ChemCoTBench: strongest gains on open-ended tasks and comparable performance on closed-ended ones. Lower “All” scores mainly reflect different open-/closed-ended task mixtures.
> |Benchmark|Overlap with ChemCoTDataset|
> |-|-:|
> |ChemCoTBench|8 records(0.71%)|
> |Other|0|
>
> **Key Q3: reporting wall-clock time**
>
> We additionally measured wall-clock time and observed a 5.96× overall speedup; please see response to Reviewer Ehya W1 / Key Q(i) for details.
>
> **Limitation (2): dual-use risk**
>
> We agree and will strengthen the limitations/impact discussion on dual-use risk in the revision.

---

> > ### Author Rebuttal · Reviewer_5Cfh · 2026-04-06
> >
> > Thanks for the detail response. The authors' rebuttal addresses my concerns. I hope they will incorporate these suggestions into the final version to further improve the paper's quality!

---

> > > ### Author Response · Authors · 2026-04-07
> > >
> > > Thank you for the feedback. We are very glad that our rebuttal addressed your concerns, and we truly appreciate your suggestions. We will incorporate them carefully in the revision to further improve the paper.

---

### Official Review · Reviewer_uYQs · 2026-03-09

**Soundness:** 2
**Presentation:** 3
**Significance:** 2
**Originality:** 2
**Overall Recommendation:** 4
**Confidence:** 4

**Summary:**

The paper introduces a framework that replaces explicit CoT reasoning in chemical LLMs with continuous latent thought vectors. Built with a molecular encoder, the architecture adds: (1) a Perceiver Resampler, (2) an Updater for iterative cross-attention between latent thoughts and molecular features, and (3) a latent projector feeding hidden states back as input. A four-stage protocol trains the system. The central claim is spontaneous internalization, yielding a better non-tie win rate on ChemCoTBench and token reduction across four benchmarks.

**Compliance With Llm Reviewing Policy:**

Affirmed.

**Final Justification:**

After the rebuttal, I raise my recommendation to a weak accept, as the authors provided substantial additional experiments that resolve several key concerns, particularly around evaluation variance, baseline design, and ablations. While some issues remain regarding the magnitude and interpretation of gains, the improved empirical support and clarifications make the contribution sufficiently solid to be of interest to the community.

**Key Questions For Authors:**

- **Q1 Missing baselines.** What are the results for Stage 1+4 (no CoT, no latent), Stage 1+2+3 (latent but no GRPO), and/or Qwen-3-8B SFT+GRPO (text-only)?

- **Q2 Train/test decontamination.** Was molecule-level decontamination performed between ChemCoTDataset and all four evaluation benchmarks? Are the GRPO rollout prompts the same 14k SFT samples?

- **Q3 CoT quality.** Why does Stage 1+2 underperform Stage 1 (35.50% vs 41.73%)?

- **Q4 Joint training in Stage 4.** What happens if latent modules are trained jointly with the LLM during GRPO rather than frozen?

- **Q5 Efficiency definition.** What does one "step" mean in Figure 8? Please provide wall-clock time and/or FLOPs per sample for LatentChem vs. the CoT baseline.

- **Q6 Per-task reward.** What does the correctness oracle check for each task type? Please provide per-task reward hit rates. What is the format adherence of each baseline?

- **Q7 Evaluation variance.** What is the tie-rate for the non-tie win rate? Were multiple seeds or evaluation runs performed? How many rollouts do you do per prompt? Could you provide any variance measure?

**Limitations:**

The authors discuss computational cost, interpretability trade-offs, and domain scope limitations. However, they do not address: (1) the significant per-task regressions visible in disaggregated results (e.g., ligand selection collapsing from 47% to 13%), (2) the sensitivity of results to reward design choices across heterogeneous tasks, or (3) the lack of evidence for generalization beyond the training distribution. Train/test decontamination are not documented.

**Strengths And Weaknesses:**

**Strengths:**

- **S1 Good Experiments** The causal ablation, budget stress test, and t-SNE dynamics are well-intended experiments that ask the right questions about what latent tokens encode. However, experiments are under-documented overall.

- **S2 Empirical results.** The molecule optimization results are impressive when comparing against other LLMs.

- **S3 Sensible architecture** The ChemUpdater — iterative cross-attention between latent thoughts and molecular features — is a well-motivated design choice.

- **S4 Benchmark coverage** The evaluation spans four benchmarks (ChemCoTBench, Mol-Instructions, ChEBI-20, ChemLLMBench). While most of these cover similar tasks, still good to have more than one result to confirm performance.

- **S5 Reproducibility.** Code and model weights are publicly available.

**Weaknesses:**

- **W1 The "spontaneous internalization" claim is undermined by the training design.** The paper's central narrative is that the model voluntarily abandons textual CoT during RL, discovering latent reasoning on its own. However, the four-stage training pipeline is specifically engineered to produce this outcome: Stage 3 trains the latent modules to reproduce CoT-equivalent representations — directly pre-loading reasoning patterns into the latent tokens. Stage 4 reverses the freeze: the latent modules are locked while the LLM is fine-tuned with GRPO using only answer correctness rewards (β=0, no KL penalty). So this is not spontaneous at all but made by design. Furthermore, freezing the latent modules in Stage 4 means: (a) they are permanently locked to their CoT-supervised initialization and never receive task-level reward signal — if the LLM-distilled CoT data was noisy, the latent tokens inherit those limitations; (b) the LLM-latent interface is never jointly optimized — Stage 3 fits latent modules to a frozen LLM, then Stage 4 changes the LLM via LoRA while the latent modules cannot adapt. Joint training of both in Stage 4 is the natural alternative and is not tested.

- **W2 Causal ablation (Fig 5) does not support "monotonic degradation" for optimization.** The optimization panel shows roughly flat, noisy performance from k=0 through k=7-8, with some lines increasing at intermediate k. The sharp drop occurs only at k=9-10 when nearly all tokens are noise — trivially expected. This pattern suggests distributed/redundant encoding, not sequential critical precursors. The understanding panel is somewhat clearer but still noisy. No error bars are reported despite stochastic generation (temp=1.5, top-p=0.9) and limited per-task samples.

- **W3 Budget stress test (Fig 6) shows CoT length, not performance.** Figure 6 only shows that the model generates more text as the latent budget shrinks. The actual performance data (Figure 11) is in the appendix. Figure 11 shows that at T=0 (pure CoT, no latent), the model still achieves ~40-80% optimization SR. The paper never compares LatentChem at T=0 to the Stage 1+2 CoT baseline.

- **W4 Evaluation metric** The non-tie win rate discards all ties, which are never reported. A 59.88% non-tie win rate with a high tie rate could reflect a marginal total-sample advantage. Standard absolute metrics would be more informative. Also, multiple runs over different seeds or at least multiple evaluations to get a sense of error.

- **W5 Negative Results buried in the appendix** Table 5: LatentChem loses on forward reaction Top-1 (0.17 vs 0.20), FTS (0.52 vs 0.63), by-product Top-1 (0.08 vs 0.12), FG counting MAE (0.09 vs 0.07), edit-delete (84% vs 88%). Table 6: ligand selection collapses from 47% to 13%. These regressions are buried in the appendix.

- **W6 GRPO is insufficiently specified for multi-task training.** The paper does not specify task mixing strategy, per-task reward hit rates, or any reward calibration across the 22+ subtasks. Only 1 epoch over 14k samples yields an extremely light RL phase that may explain the inconsistent per-task results.

- **W7 Efficiency claim lacks wall-clock validation.** The token count ratio is interesting, but never really defines what #steps means for each of the different models. Also, each latent token costs more compute (LLM forward pass + ChemUpdater + projector) than a text token. Actual throughput is never measured. Additional metrics would be appreciated, like wall clock time and flops.

- **W8 Baseline design confounds multiple factors.** The comparison between Stage 1+2+4 (35.50%) and LatentChem (59.88%) is supposed to isolate the latent modules, but the training paths differ (LatentChem passes through Stage 3, Stage 1+2+4 does not). Key missing ablations: Stage 1+4 (GRPO without CoT or latent), Stage 1+2+3 (latent without GRPO), and Qwen-3-8B SFT+GRPO (text-only). Without a clean 2x2 (latent yes/no x GRPO yes/no), individual contributions are confounded. The Table 4 component ablation is also confounded — components are removed from a jointly-trained system rather than training from scratch without them.

- **W9 t-SNE dynamics (Fig 7a) undermine the case for 10 latent steps.** ChemTokens disentangle into task-specific clusters within steps 1-2 and remain stable from steps 3-10. If representations don't change after step 2, steps 3-10 appear redundant — consistent with the Figure 5 observation that corrupting later tokens has minimal effect.

- **W10 Training pipeline underdocumented.** No train/test decontamination is reported between ChemCoTDataset and the four evaluation benchmarks (all draw from overlapping public databases). Per-task distribution of the 14k training samples is unspecified. GRPO appears to reuse the same SFT samples, limiting generalization claims. Coconut-Chem reproduction details (training stages, data, hyperparameters) are absent.

- **W11 Counterfactual alignment never ablated.** Stages 1 and 2 employ a hinge loss that penalizes the model when corrupted ChemTokens produce a similar loss to clean ChemTokens. This is intended to force the model to rely on molecular information rather than textual priors. The mechanism is unusual — standard chemical LLM adapter papers (MolLLaMA) do not use it — and its contribution is never isolated. Without an ablation comparing Stage 1 with vs. without L_CF, we cannot tell whether it is essential or unnecessary complexity.

---

> ### Author Rebuttal · Authors · 2026-03-31
>
> **Fig&table link**: https://github.com/anonymous-rebuttal2026/anonymous
>
> **W1/Q4: “spontaneous internalization” and joint Stage-4 training**
>
> We appreciate this point. We do not claim the model “discovers” latent reasoning from scratch: Stage 3 provides the latent interface, and the question is which channel the policy chooses once both latent and textual channels are available. Because “spontaneous internalization” was not explicitly defined, we will define it at first use as “self-selected under outcome-only optimization” in revision. Joint Stage-4 training by unfreezing the latent modules with the LLM is worse than our frozen-latent design in both overall performance and token efficiency(see details in `stage4-joint-train.pdf in link↑`).
>
> **W2/W9: Fig. 5 monotonicity & error bar requirement; potentially redundant later latent steps**
>
> Yes, Fig. 5 should be described as showing an overall degradation trend, not monotonicity. We added error bars(±1 standard error; see `error-bar.pdf in link↑`), and the trend is clearer. The results suggest later latent steps are not equally necessary.
>
> **W3: budget stress test should show performance; T=0 vs CoT**
>
> We will move Fig. 11 to the main text. We compared T=0 vs CoT baseline(see details in `budget0-vs-cot.pdf in link↑`), T=0 is largely comparable to CoT on optimization, indicating fallback to textual reasoning, though some topology-sensitive understanding tasks still benefit from latent budget.
>
> **W4/Q7: evaluation metric & variance**
>
> We now report win/lose/tie rates for all Table 2 comparisons(see `win-lose-tie.pdf in link↑`) and do not observe tie patterns changing the conclusions. We also repeated evaluation over 50 seeds(2026–2075); mean ± std non-tie win rates remain stable across benchmarks and consistent with table 2 in the manuscript. See details in `robust-result.pdf in link↑`.
>
> **W5: negative results**
>
> We agree some closed-ended subtasks regress; our claim is not per-subtask dominance, and we will discuss these results more clearly in main text. Details in response to Reviewer Ehya W2/Key Q(ii).
>
> **W6/Q6: GRPO details, reward hit rates, format adherence**
>
> GRPO samples ChemCoTDataset tasks uniformly, with correctness checked by task-specific benchmark oracles(e.g., property evaluators); per-task reward hit rates are in `reward-hit-rate.pdf in link↑`. Except the untuned text-only baseline, most format adherence is >97%(see `format-adherence-rate.pdf in link↑`). We do not use task-specific reward calibration. We trained a model with 3 GRPO epochs. Results(see `stage4-3epochs.pdf in link↑`) show open-ended performance drops slightly, closed-ended performance rises slightly, and token efficiency improves. All these will be incorporated in revision.
>
> **W7/Q5: “step” meaning, wall-clock**
>
> In Fig. 8, one step means generating one latent token or one text token. We measured wall-clock time and observed a 5.96× overall speedup; see response to Reviewer Ehya W1/Key Q(i) for details.
>
> **W8/Q1: baseline design confounds multiple factors**
>
> Stage 1+2+4 is the explicit-CoT baseline and contains no latent modules, and Table 4 reports from-scratch ablations(will be clarified in revision). We additionally trained Stage 1+4, Stage 1+2+3, and Qwen-3-8B(SFT+GRPO), all evaluated with the same Table-2 non-tie win rate against the CoT baseline(see `additional-baselines.pdf in link↑`). GRPO alone and latent without GRPO are weaker, while LatentChem remains strongest overall across all four benchmarks.
>
> **W10/Q2: training pipeline underdocumented**
>
> For decontamination, please refer to response to Reviewer 5Cfh Q2. The per-task distribution follows the official ChemCoTDataset release. GRPO rollouts use the same ChemCoTDataset prompt pool as SFT, so the generalization evidence is transfer to disjoint evaluation benchmarks, not unseen RL prompts; separate RL data could strengthen this and will be noted as a limitation. Coconut-Chem uses same data and hyperparameters as LatentChem, is first trained with our stage 1&2, then trained with Coconut’s official latent-reasoning procedure. All these will be incorporated in revision.
>
> **W11: Counterfactual alignment never ablated**
>
> We added a direct Stage-1 ablation of $L_{CF}$(see non-tie win rate in `cf-loss-ablation.pdf in link↑`). The results suggest $L_{CF}$ is not essential for main gains; a simpler Stage-1 recipe without it remains competitive and is slightly better overall here. We will position it as an optional grounding regularizer.
>
> **Q3: CoT quality**
>
> We believe this mainly reflects the cost of explicit CoT imitation. Stage 1 optimizes the final answer directly, while Stage 2 additionally forces imitation of a single reference rationale, which can overconstrain tasks with many valid intermediate paths and shift too much capacity toward reproducing linguistic traces rather than final-answer quality.
>
> **Limitation**
>
> We will revise the limitations section to explicitly address the above-mentioned points.

---

> > ### Author Rebuttal · Reviewer_uYQs · 2026-04-02
> >
> > I thank the authors for the rebuttal. The additional experiments represent a significant effort and address several of my original concerns. In particular, W4 (evaluation variance), W8 (confounded baselines), W10 (decontamination), and W11 (CF loss ablation) are adequately addressed.
> >
> > However, some concerns remain, and the new evidence raises follow-up questions:
> >
> > 1. On ChemCoTBench, ties account for 55% of comparisons. The actual win/lose split is 27.0% vs 18.0% — a real but modest advantage. On Mol-Instructions "All," wins and losses are nearly equal (32.0% vs 32.2%). I would ask the authors to report absolute task-specific metrics (success rates, accuracy) as the primary results in Table 2, with the pairwise win rate as a supplement. This would give readers a much clearer picture of the practical magnitude of improvements.
> >
> > 2. The new baselines show that every SFT stage (Stage 2 and Stage 3) degrades performance relative to Stage 1, and GRPO is what recovers it. Stage 1+2+3 scores only 33.33% on ChemCoTBench — worse than the CoT baseline. The cleanest comparison isolating the latent contribution is Stage 1+4 (47.83%) vs. LatentChem (59.88%). Could the authors comment on this ~12-point gap as the marginal contribution of latent tokens, and whether the current framing (latent reasoning as the primary contribution) is proportionate, given that GRPO appears to be doing much of the heavy lifting?
> >
> > 3. Open-ended performance drops from 59.88% to 49.31% on ChemCoTBench with additional GRPO training. This suggests the reported result may reflect a sensitive training sweet spot rather than a stable optimum. Could the authors comment on the robustness of results to training duration and whether there is a principled way to select the stopping point?
> >
> > 4. The joint-training experiment shows both lower performance and a mol.optimization efficiency ratio of 0.83× (slower than baseline). While this validates the frozen-latent design choice, it also suggests the latent representations from Stage 3 are brittle and cannot withstand further gradient updates. This is difficult to reconcile with the claim that these representations encode robust chemical logic.

---

> > > ### Author Response · Authors · 2026-04-03
> > >
> > > Thank you for the thoughtful follow-up. We address the main points below in turn.
> > >
> > > 1. We agree that absolute task-specific metrics should be the primary presentation in Table 2, with pairwise win rates reported as a supplement. We will revise the table accordingly; this change does not alter the paper’s conclusions.
> > >
> > > 2. We thank the reviewer for raising this point. We would like to clarify that this is not a comparison between latent reasoning and GRPO, since they play different roles: latent thinking is the reasoning interface, whereas GRPO is the training method. The 47.83% to 59.88% gain is the marginal contribution of the latent interface on top of the same GRPO training. We will revise the framing to make this distinction clearer.
> > >
> > > 3. We appreciate this question. According to prior work on reinforcement-learning-based post-training, performance can be non-monotonic with training duration, and additional optimization does not necessarily lead to better task performance [1,2]. We therefore treat 1 epoch as a conservative practical stopping point in this work, which is empirically effective in our setting. We also acknowledge that GRPO itself can exhibit training instability and over-optimization, and we will incorporate this discussion into the revision.
> > >
> > > 4. We thank the reviewer for this point. We do not interpret the joint-training result as evidence that the learned latent representations are brittle. We instead interpret this result primarily as an optimization issue: GRPO itself can be unstable, and the latent thinking modules are lightweight relative to the backbone, making joint training harder to optimize well. This is consistent with our staged design, which first learns the latent interface with a frozen backbone and then keeps it fixed during RL. We will revise the wording accordingly.
> > >
> > > [1] Gao, Leo, John Schulman, and Jacob Hilton. "Scaling laws for reward model overoptimization." International Conference on Machine Learning. PMLR, 2023.
> > >
> > > [2] Zhu, Banghua, Michael I. Jordan, and Jiantao Jiao. "Iterative data smoothing: Mitigating reward overfitting and overoptimization in rlhf." arXiv preprint arXiv:2401.16335 (2024).

---

### Decision · Program_Chairs · 2026-04-30

**Decision:**

Accept (regular)

**Comment:**

This paper proposes LatentChem, a framework that replaces explicit chain-of-thought reasoning in chemical LLMs with continuous latent thought vectors, achieving notable gains on open-ended chemical tasks and substantial reductions in reasoning token overhead. All three reviewers initially gave weak accept, raising concerns about the "spontaneous internalization" framing, overstated efficiency claims, mixed closed-ended results, and insufficient ablations, though the authors addressed most of these through additional experiments including wall-clock measurements, new baselines, and topology-sensitive ablations, leading to maintained or improved scores after rebuttal. I recommend acceptance contingent on the authors toning down the stronger claims around spontaneous internalization and clearly separating token-overhead reduction from actual runtime efficiency in the final version.